 # Are Tools Always Beneficial? Learning to Invoke Tools Adaptively for Dual-Mode Multimodal LLM Reasoning

**Qinghe Ma** [1 2 *]  **Zhen Zhao** [2]  **Yiming Wu** [3]  **Jian Zhang** [1]  **Lei Bai** [2]  **Yinghuan Shi** [1 4]

## Abstract

Tool-augmented reasoning has emerged as a promising direction for enhancing the reasoning capabilities of multimodal large language models (MLLMs). However, existing studies mainly focus on enabling models to perform tool invocation, while neglecting the necessity of invoking tools. We argue that tool usage is not always beneficial, as redundant or inappropriate invocations largely increase reasoning overhead and even mislead model predictions. To address this issue, we introduce AutoTool, a model that adaptively decides whether to invoke tools according to the characteristics of each query. Within a reinforcement learning framework, we design an explicit dual-mode reasoning strategy with mode-specific reward functions to guide the model toward producing accurate responses. Moreover, to prevent premature bias toward a single reasoning mode, AutoTool jointly explores and balances tool-assisted and text-centric reasoning throughout training, and promotes free exploration in later stages. Extensive experiments demonstrate that AutoTool exhibits outstanding performance and high efficiency, yielding a 21.8% accuracy gain on V* benchmark compared to the base model, and a 44.9% improvement in efficiency over existing tool-augmented methods on POPE benchmark. Code is available at https://github.com/MQinghe/AutoTool.

---

*This work was done during the internship at Shanghai Artificial Intelligence Laboratory. [1]The State Key Laboratory for Novel Software Technology, Nanjing University, Nanjing, China [2]Shanghai Artificial Intelligence Laboratory, Shanghai, China [3]School of Computing and Data Science, The University of Hong Kong, Hong Kong, China [4]The Institute of Brain-Computer Interface, Nanjing University, Nanjing, China. Correspondence to: Zhen Zhao <zhaozhen@pjlab.org.cn>, Yinghuan Shi <syh@nju.edu.cn>.

*Proceedings of the 43rd International Conference on Machine Learning*, Seoul, South Korea. PMLR 306, 2026. Copyright 2026 by the author(s).

## 1. Introduction

By decomposing complex problems into a sequence of reasoning steps, chain-of-thought (CoT) prompting (Wei et al., 2022; Kojima et al., 2022) has endowed multimodal large language models (MLLMs) (Team et al., 2023; Liu et al., 2023; Wang et al., 2024b; Bai et al., 2025) with stronger reasoning capabilities. However, most existing approaches follow the textual reasoning paradigm of large language models (LLMs) (Achiam et al., 2023; Dubey et al., 2024; Guo et al., 2025), leaving current MLLMs constrained by linguistic bias that limits their ability to effectively leverage multimodal information. The multimodal CoT (MCoT) prompt (Zhang et al., 2025; Wang et al., 2025a), exemplified by the "Thinking with Images" approach of OpenAI o3 (OpenAI, 2025), injects multimodal context into reasoning to strengthen visual cues and cross-modal interactions.

In MCoT, visual information is typically derived from external tools such as additional search engines (Fan et al., 2024; Komeili et al., 2021), multiple visual models (Ma et al., 2025c; Qi et al., 2024), or image processing methods (Su et al., 2025b; Zheng et al., 2025). Recent progress in reinforcement learning (Shao et al., 2024; Guo et al., 2025; Chen et al., 2025) allows models to acquire tool-usage skills in a more cost-efficient and flexible way (Su et al., 2025b; Zheng et al., 2025; Su et al., 2025a). While MCoT demonstrates superior reasoning capabilities compared to text-centric CoT on multiple benchmarks, it also introduces two major challenges. **The first lies in the significantly increased training and inference costs.** Existing tool-augmented reasoning models, such as OpenThinkIMG (Su et al., 2025b) and DeepEyes (Zheng et al., 2025), often rely on fixed tool invocation orchestration or inadequate reward designs. Consequently, they **implicitly** focus on learning how to invoke tools correctly and generate accurate answers, while neglecting whether tool usage is truly necessary. As illustrated in Figure 1(c) and Figure 1(d), taking DeepEyes (Zheng et al., 2025) as an example, it consistently encourages tool invocation regardless of task difficulties. Even for simple queries, the model tends to engage in unnecessary multi-turn reasoning, substantially increasing computational overhead during both training and inference. Hence, DeepEyes requires 44.9 training hours,

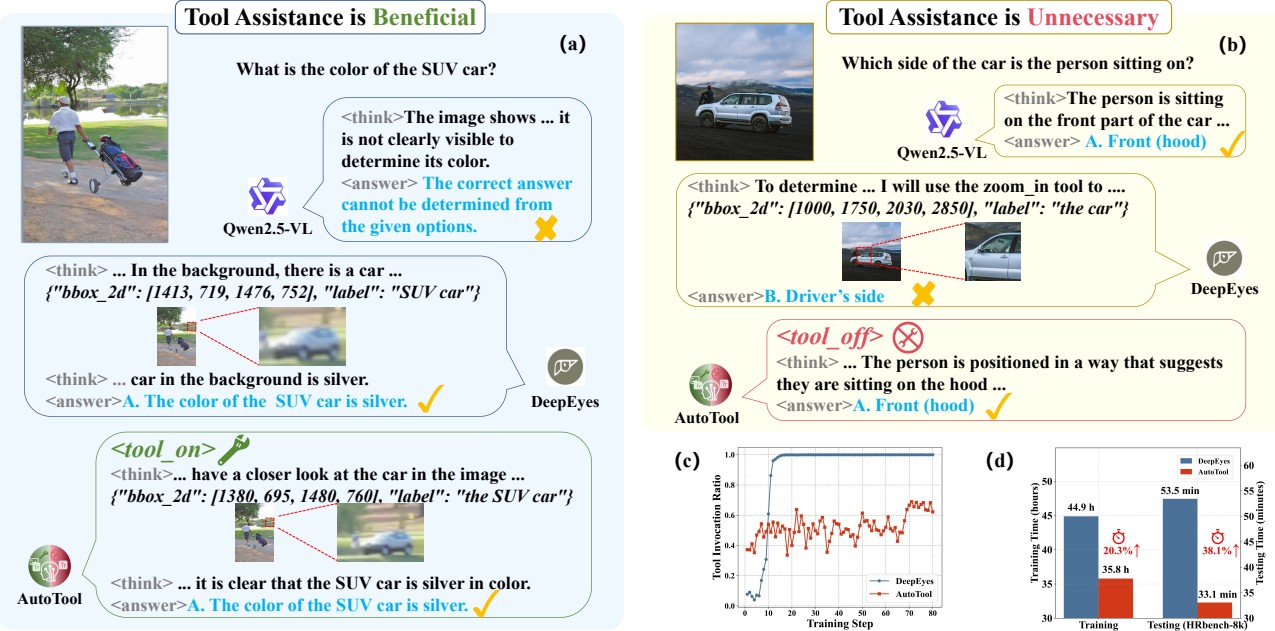

Figure 1. (a, b) Representative queries that do or do not trigger the zoom-in tool, illustrating that tool usage is not always necessary, while AutoTool adaptively invokes tools when beneficial. (c, d) Comparison of the proportion of tool-augmented reasoning trajectories during training, as well as the training and inference time costs between our AutoTool and SOTA DeepEyes (Zheng et al., 2025).

20.3% more than adaptive tool invocation, indicating that redundant tool usage severely slows down the reasoning process. **Furthermore, erroneous tool invocations may interfere with reasoning.** As shown in Figure 1(b), when answering a question about the spatial relationship between a person and a car, the model should rely on global understanding, where zoom-in tool invocation is unnecessary. However, DeepEyes incorrectly invokes the tool to focus solely on the car region, rather than the combined area of the car and the person, introducing redundant visual information that distracts the reasoning process and ultimately leads to hallucinated responses. In such cases, the autoregressive nature of LLMs makes frequent tool invocations particularly problematic, as they amplify irrelevant visual cues and cause error accumulation, further intensifying reasoning distraction and hallucination.

In our opinion, when handling a multimodal query, an ideal model should carefully *determine whether tool assistance is necessary* before invocation. Taking the zoom-in operation as an example, intuitively, if a question requires close inspection or verification of fine-grained visual details, the zoom-in tool becomes essential. As illustrated in Figure 1(a), where the task involves identifying a specific object among multiple candidates, zooming into the target region substantially improves the likelihood of a correct answer. In contrast, as shown in Figure 1(b), when the question involves global understanding, overall layout reasoning, or when the target region is already sufficiently clear, invoking the zoom-in tool yields negligible benefit and may even

introduce unnecessary distractions.

To address the issue of existing methods that overemphasize tool usage, we introduce *AutoTool*, which empowers the model to adaptively decide when to engage in "Think with Images" reasoning, reconsidering the common belief that "tools are always beneficial". By **explicitly** controlling tool usage through two special tokens, `<tool_on>` and `<tool_off>`, AutoTool employs dual reasoning modes that leverage tools for complex problems while recognizing that simple queries can be solved without tool assistance. This paradigm improves both training and inference efficiency, as well as mitigating hallucinated responses. Instead of relying on carefully curated SFT data for cold-start training, we adopt an end-to-end reinforcement learning framework that encourages the model to fully explore the two reasoning modes in a simple yet effective manner.

Within this explicit dual-mode paradigm, we design distinct reward functions to evaluate reasoning trajectories under different reasoning modes, which we refer to as **Mode-Specific Policy Optimization (MSPO)**. For the `<tool_on>` mode, the model is trained to accurately utilize the tool while providing correct answers. Unlike prior methods (Su et al., 2025b; Zheng et al., 2025) that primarily emphasize tool invocation, we penalize instances where the model invokes tools but produces incorrect answers, reducing unnecessary or ineffective tool operations. For the `<tool_off>` mode, the model relies entirely on its internal reasoning to generate accurate answers. However, learning to master dual reasoning modes is nontrivial. Due to the inherent rea-

soning bias of the foundation model, the policy model tends to prefer the `<tool_off>` mode, which often yields higher rewards more easily, leaving the `<tool_on>` mode underexplored. To mitigate this imbalance, we propose an **Adaptive Mode Balancing (AMB)** strategy that dynamically adjusts the reward coefficients to control the frequency of the two modes, ensuring sufficient exploration for both. The constraint is relaxed in the later stage of training, allowing the model to freely determine its preferred mode. Our contributions can be summarized as follows:

- We analyze the pros and cons of tool-assisted reasoning for MLLMs. While tool invocation can enhance reasoning capabilities, blindly encouraging tool usage increases both training and inference costs and may introduce distracting or redundant information.

- We design Mode-Specific Policy Optimization (MSPO), with distinct optimization objectives to different reasoning modes, enabling the model to learn adaptive reasoning with or without tools.

- We propose Adaptive Mode Balancing (AMB), which adaptively and dynamically adjusts the frequency of the two modes to ensure sufficient exploration of dual reasoning modes throughout training.

Extensive experiments on multiple multimodal benchmarks demonstrate that AutoTool achieves superior reasoning capability and high efficiency.

## 2. Related Work

### 2.1. Multimodal Large Language Models

The emergence of multimodal large language models (MLLMs) (Liu et al., 2023; Team et al., 2023; Hurst et al., 2024; Bai et al., 2025; Fa et al., 2026; Li et al., 2026) marks a major milestone in artificial intelligence and has substantially promoted the development of diverse application domains (Ma et al., 2024; 2025a;b; Duan et al., 2025; Yang et al., 2025; Wang et al., 2025d). Early works such as LLaVA (Liu et al., 2023; 2024), BLIP (Li et al., 2022; 2023a), and Qwen-VL (Bai et al., 2023; Wang et al., 2024b; Bai et al., 2025) adopt modular architectures that pair pretrained visual encoders (*e.g.*, CLIP-ViT (Cherti et al., 2023; Radford et al., 2021), InternViT (Chen et al., 2024)) with LLMs, laying the foundation for MLLM development. These models typically involve large-scale multimodal alignment training followed by instruction tuning for task adaptation. Subsequent studies like Flamingo (Alayrac et al., 2022) and Cambrian-1 (Tong et al., 2024) integrate multiple encoders for richer visual representations, while EVE (Diao et al., 2024), MonoInternVL (Luo et al., 2025), and SAIL (Lei et al., 2025) pursue end-to-end architectures

that process raw image patches and text tokens within a unified Transformer. Recently, reinforcement learning has further advanced chain-of-thought (CoT) reasoning (Shao et al., 2024; Guo et al., 2025; Chen et al., 2025), yet most approaches remain text-centric (Fan et al., 2025; Yao et al., 2025), limiting the model's understanding of visual content. To address this, we propose adaptive tool-assisted zoom-in reasoning for complex problems, enabling deeper visual exploitation and more interpretable answers.

### 2.2. Tool-Augmented Reasoning in MLLMs

The multimodal information processing capability of MLLMs enables human-like "Thinking with Images" through multimodal chain of thought (MCoT) reasoning (Zhang et al., 2025; Wang et al., 2025a; Su et al., 2025c; Zheng et al., 2025; OpenAI, 2025). Recent works such as Visual Sketchpad (Hu et al., 2024), OpenThinkIMG (Su et al., 2025b), and Thyme (Zhang et al., 2025) equip models with planning and orchestration abilities, leveraging diverse external tools, such as semantic segmentation (Kirillov et al., 2023; Ravi et al., 2024), OCR, and depth estimation (Yang et al., 2024b;c), to inject rich visual cues into the reasoning process. Beyond explicit tool usage, methods like BAGEL (Deng et al., 2025), Visual Planning (Xu et al., 2025), and GoT (Fang et al., 2025) unify generation and reasoning, generating new explicit or implicit visual states from contextual semantics to facilitate subsequent reasoning steps. Current approaches for acquiring tool-use capability typically fall into three categories: prompt-based methods that rely on in-context learning (Hu et al., 2024; Li et al., 2025), supervised fine-tuning that teaches procedural competence from examples (Wu & Xie, 2024; Ma et al., 2025c), and reinforcement learning that optimizes tool-use policies through feedback (Su et al., 2025b; Lai et al., 2025). However, existing studies mainly emphasize how to teach models to use tools correctly, neglecting the critical question of whether tool invocation is necessary. Thus, our method adaptively decides when and how to invoke tools, achieving a balance between reasoning efficiency and reliability.

### 2.3. Reinforcement Learning in Large Models

Reinforcement learning has demonstrated remarkable potential in enhancing the reasoning capabilities of large models (Shao et al., 2024; Guo et al., 2025; Chen et al., 2025). DeepSeek-R1 (Guo et al., 2025) shows that even simple rule-based RL strategies can effectively induce strong reasoning behaviors, inspiring a surge of research into RL-based reasoning enhancement. Building on this trend, recent works such as DeepEyes (Zheng et al., 2025), TreeVGR (Wang et al., 2025a), and Thyme (Zhang et al., 2025) employ Group Relative Policy Optimization (GRPO) (Shao et al., 2024) to guide models in performing accurate tool-assisted reasoning. Distinct from these approaches, our method leverages

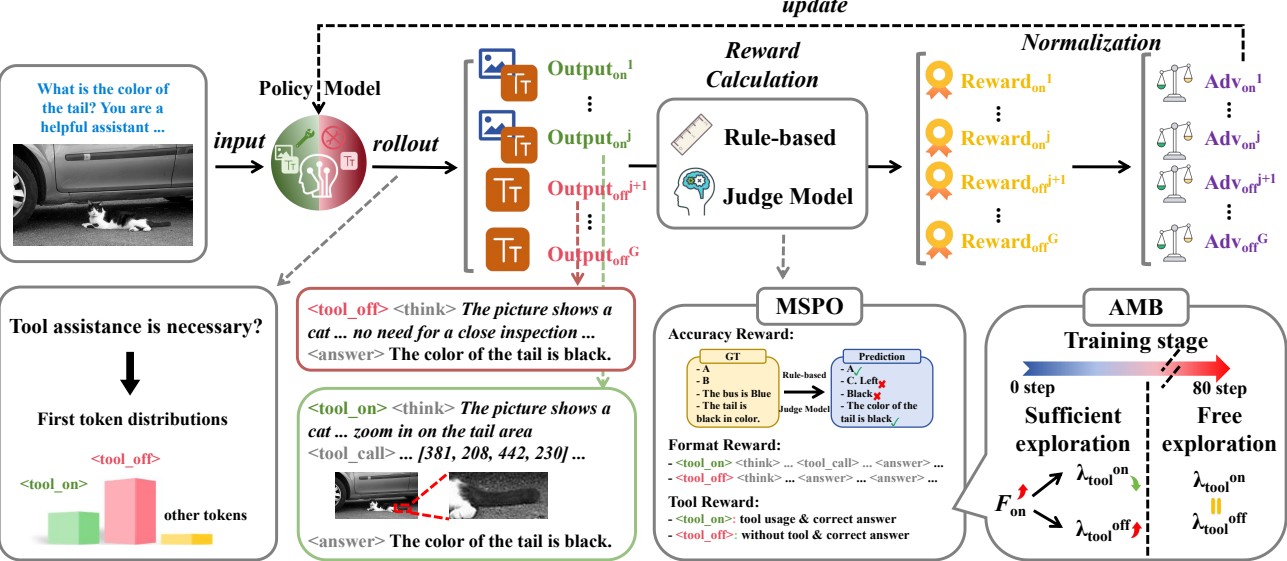

*Figure 2.* Illustration of the AutoTool training framework. Given a multimodal problem, the policy model first decides whether the subsequent reasoning process requires tool invocation. For each batch of generated reasoning trajectories, different reward functions are applied to evaluate the trajectories under distinct reasoning modes via the Mode-Specific Policy Optimization (MSPO), and the tool invocation reward coefficient is estimated through the Adaptive Mode Balancing (AMB) strategy. The model is optimized via the GRPO.

RL not only to reinforce proper tool invocation, but also to explore and coordinate multiple reasoning modes, fostering more adaptive and context-aware multimodal reasoning.

## 3. Method

### 3.1. Problem Formulation and Preliminary

Given a multimodal query $X = (Q, V)$, where $Q$ denotes the textual query and $V$ represents the visual inputs, we first revisit the traditional **text-centric reasoning paradigm**. In this paradigm, the policy model $\pi_\theta$ performs reasoning purely in the textual space by generating a sequence of intermediate reasoning steps $R = \{T_i\}_{i=1}^I$, where $T_i$ represents the internal reasoning text at the $i$-th step. At each step, the reasoning state $R_i \sim \pi_\theta(\cdot \mid X, R_1, \ldots, R_{i-1})$ is sampled from $\pi_\theta$ conditioned on the initial query and all previous steps. Each new reasoning step is appended to the context and fed back into the policy model for subsequent reasoning. This iterative process continues until the model outputs the final answer $Y$, or until a predefined limit on context length is reached. Accordingly, the complete textual reasoning trajectory $\gamma_t$ can be formulated as:

$$\gamma_t = \{X, (T_1), \ldots, (T_I, Y)\}. \quad (1)$$

Different from the text-only paradigm, **multimodal reasoning paradigm** augments each step with tool interactions. Specifically, the reasoning state at the $i$-th step is represented as a triplet $R_i = (T_i, A_i, O_i)$, where $T_i$ denotes the internal reasoning text, $A_i$ denotes the tool action along with its parameters, and $O_i$ denotes the observation returned by executing the tool action. The complete multimodal reasoning

trajectory $\gamma_m$ alternates between reasoning and interaction, and terminates with a final textual answer $Y$. Accordingly, the resulting multimodal reasoning trajectory is defined as:

$$\gamma_m = \{X, (T_i, A_i, O_i)_{i=1}^{I-1}, (T_I, Y)\}. \quad (2)$$

### 3.2. Overview

Compared with the text-centric reasoning paradigm, multimodal reasoning extends visual information processing from a one-time encoding to an iterative editing process through explicit tool invocation. This paradigm allows the model to step out of the textual bias and effectively leverage multimodal cues for reasoning. However, indiscriminately encouraging the model to invoke tools leads to two major issues: (1) the reasoning cost increases significantly during both training and inference, and (2) unnecessary or incorrect tool usage may introduce noisy or misleading information, thereby deteriorating the reasoning reliability.

We introduce **AutoTool** to break the conventional assumption that "tools are always beneficial" in multimodal reasoning. It adaptively decides whether tool invocation is necessary for each task and selects the more suitable reasoning mode, achieving a better balance between reasoning efficiency and answer reliability. As illustrated in Figure 2, given a user-provided multimodal query, AutoTool first determines whether the current question requires the assistance of the tool. If tool usage is deemed necessary, the policy model invokes a zoom-in function to locate the region of interest that is most relevant to the query, and appends the resulting cropped visual observation to the reasoning context

for subsequent inference. Otherwise, the policy model performs purely textual reasoning to directly produce the final answer in a more efficient manner. The policy for whether and how to invoke tools is learned through reinforcement learning, as detailed in the following sections.

### 3.3. Explicit Dual Reasoning Modes

We define two special control tokens, `<tool_on>` and `<tool_off>`, explicitly indicating whether the model employs tools in subsequent reasoning. `<tool_on>` triggers tool-augmented reasoning with `<tool_call>` and `<tool_response>` structures, while `<tool_off>` corresponds to pure textual reasoning without tool usage. We carefully design the prompts, as detailed in the supplementary material, to explicitly define the applicable scenarios and output formats for both reasoning modes. The policy model is trained via Group Relative Policy Optimization (GRPO) (Shao et al., 2024), a reinforcement learning algorithm that enables effective and efficient exploration of different reasoning strategies without relying on hard-to-obtain SFT data.

Specifically, given a multimodal query $X = (Q, V)$, we sample a group of $G$ candidate reasoning trajectories $\{o_i\}_{i=1}^{G}$ from the policy model. For each trajectory $o_i$ from the old policy $\pi_{\theta_{old}}$, we compute a scalar reward $r_i$ based on both the final answer and the intermediate reasoning process, as detailed in Section 3.4. The rewards $\{r_i\}_{i=1}^{G}$ are then normalized to obtain the advantages $\{\hat{A}_i\}_{i=1}^{G}$. Formally, the optimization objective of GRPO is defined as:

$$\mathcal{J}_{\text{GRPO}}(\theta) = \mathbb{E}_{X, \{o_i\}_{i=1}^{G} \sim \pi_{\theta_{\text{old}}}} \left[ \frac{1}{G} \sum_{i=1}^{G} \min \left( \frac{\pi_\theta(o_i|X)}{\pi_{\theta_{\text{old}}}(o_i|X)} \hat{A}_i, \right. \right.$$
$$\left. \left. \text{clip} \left( \frac{\pi_\theta(o_i|X)}{\pi_{\theta_{\text{old}}}(o_i|X)}, 1 - \epsilon, 1 + \epsilon \right) \hat{A}_i \right) \right],$$
(3)

$$\hat{A}_i = \frac{r_i - \text{mean}(\{r_1, r_2, \dots, r_G\})}{\text{std}(\{r_1, r_2, \dots, r_G\})},$$
(4)

where $\epsilon$ is the clipping hyperparameter and we do not include a KL regularization term.

Nevertheless, due to the intrinsic reasoning bias inherited from the foundation model, the policy model exhibits a tendency to over-prefer the `<tool_off>` mode, which yields higher rewards with less effort and consequently hinders adequate exploration of the `<tool_on>` mode. To encourage sufficient exploration across both reasoning modes, we propose the **Adaptive Mode Balancing (AMB)** strategy that dynamically regulates their respective reward coefficients, ensuring that neither mode is neglected during training.

For a batch of $N$ samples $\{X_i\}_{i=1}^{N}$, we obtain $N \times G$

rollouts from different reasoning modes. We record the occurrence counts of the two modes as $N_{\text{on}}$ and $N_{\text{off}}$, respectively, and compute the tool invocation frequency as $F_{\text{on}} = \frac{N_{\text{on}}}{N_{\text{on}} + N_{\text{off}}}$. Based on the initial tool invocation reward coefficient $\lambda_{\text{tool}}^{\text{base}}$, we dynamically adjust it as

$$\lambda_{\text{tool}}^{\text{mode}} = \begin{cases} \lambda_{\text{tool}}^{\text{base}} + 0.5 - F_{\text{on}}, & \text{if mode = on,} \\ \lambda_{\text{tool}}^{\text{base}} - 0.5 + F_{\text{on}}, & \text{if mode = off,} \end{cases}$$
(5)

where $\lambda_{\text{tool}}^{\text{mode}}$ denotes the adaptive tool invocation reward coefficient, determined by the reasoning mode of the trajectory. When the tool invocation frequency becomes too high, $\lambda_{\text{tool}}^{\text{on}}$ decreases while $\lambda_{\text{tool}}^{\text{off}}$ increases, encouraging the model to explore the `<tool_off>` mode more actively, and vice versa. Through adaptive adjustment, the model is encouraged to sufficiently explore both modes during training.

As training progresses, the model becomes proficient in both reasoning modes. At the final stage of training (*e.g.*, the last 20 steps), we remove this adaptive constraint and set $\lambda_{\text{tool}}^{\text{on}} = \lambda_{\text{tool}}^{\text{off}} = \lambda_{\text{tool}}^{\text{base}}$, allowing the policy to autonomously determine which reasoning mode to employ for each query based on its internal confidence and problem characteristics. This transition enables the model to shift from guided exploration to self-directed reasoning, achieving a more natural integration of both reasoning paradigms.

### 3.4. Mode-Specific Policy Optimization

To encourage the model to explore different reasoning modes through reinforcement learning while ensuring that it correctly follows the required output formats and performs valid tool invocations for accurate question answering, we design the following reward.

The overall reward consists of three components: accuracy reward $R_{\text{acc}}$, format compliance reward $R_{\text{format}}$, and mode-specific tool invocation reward $R_{\text{tool}}$,

$$R = R_{\text{acc}} + R_{\text{format}} + \lambda_{\text{tool}}^{\text{mode}} R_{\text{tool}}.$$
(6)

**Accuracy reward $R_{\text{acc}}$:** We evaluate whether the predicted answer is semantically equivalent to the ground truth using a combination of rule-based metrics and an online reward model (*e.g.*, Qwen2.5-72B-Instruct).

**Format reward $R_{\text{format}}$:** This ensures that the reasoning process and final answer adhere to the prescribed output format, *i.e.*, enclosed within `<think></think>` and `<answer></answer>` tags, respectively.

**Mode-specific tool reward $R_{\text{tool}}$:** For the `<tool_on>` mode, the model receives $R_{\text{tool}} = 1$ when it correctly performs the zoom-in tool invocations and produces a correct answer. If the tool is invoked but the answer is incorrect, a penalty $R_{\text{tool}} = -0.5$ is applied to account for the extra

cost of tool usage. In all other cases, $R_{\text{tool}} = 0$. For the `<tool_off>` mode, the model is rewarded $R_{\text{tool}} = 1$ only if it does not invoke the zoom-in tool and provides a correct answer; otherwise, $R_{\text{tool}} = 0$.

### 3.5. Inference

During inference, we employ the same prompting scheme as used in training. The model can autonomously select the reasoning mode based on the characteristics of the query. Alternatively, the reasoning mode can be manually specified, either by explicitly instructing the model in prompt to perform or skip tool invocation, or by appending the special token `<tool_on>` or `<tool_off>` to the input sequence.

## 4. Experiments

### 4.1. Benchmarks and Metrics

We evaluate our model across three categories of benchmarks to comprehensively assess its performance and compare it with existing methods.

**Perception benchmarks.** These include the V* (Wu & Xie, 2024) and HRbench (Wang et al., 2025b) datasets, which consist of high-resolution images (ranging from 2K to 8K). The questions in these datasets focus mainly on single-object attributes, object counting, or relative spatial relationships. The evaluation metric is the question answering accuracy.

**Grounding benchmarks.** This category includes Ref-COCO (Caesar et al., 2018), RefCOCO+ (Caesar et al., 2018), RefCOCOg (Kazemzadeh et al., 2014), and Reason-Seg (Lai et al., 2024). Both the COCO series and ReasonSeg require the model to output the bounding-box of the referred object within an image. We evaluate grounding accuracy by computing the Intersection-over-Union (IoU) between the predicted and ground-truth regions, with a threshold of 0.5 to determine whether the prediction is considered correct.

**Hallucination benchmark.** POPE (Li et al., 2023b) serves as a hallucination detection benchmark that evaluates whether the target object truly exists in the image, and its metric is the prediction accuracy.

**Reasoning benchmarks.** These include MathVista (Lu et al., 2023), MathVerse (Zhang et al., 2024), MathVision (Wang et al., 2024a), WeMath (Qiao et al., 2024), DynaMath (Zou et al., 2024), and LogicVista (Xiao et al., 2024). The tasks cover a wide range of reasoning types, including mathematical reasoning, geometric pattern recognition, logical and physical reasoning, chart interpretation, and commonsense reasoning in real-world scenarios. Some questions require the model to infer implicit information from the given text or image context. The performance metric is the accuracy of the answer.

### 4.2. Implementation Details

Following DeepEyes (Zheng et al., 2025), the training data include fine-grained samples from the V* (Wu & Xie, 2024) dataset, chart data from ArxivQA (Li et al., 2024b), and reasoning data from ThinkLite-VL (Wang et al., 2025c). The inclusion of reasoning data aims to enhance the general reasoning robustness of the model and mitigate overfitting to modality-specific patterns, where purely textual reasoning and answer generation are performed without relying on tool-based interactions. We use Qwen2.5-VL-7B (Bai et al., 2025) as the base policy model and train it with GRPO (Shao et al., 2024; Sheng et al., 2024) for 80 iterations on eight H200 GPUs. An additional two H200 GPUs are used to deploy the reward model, Qwen2.5-72B-Instruct (Yang et al., 2024a), via the vLLM (Kwon et al., 2023) inference engine. Each training batch contains 256 samples, which are divided into 4 PPO mini-batches. For each query, the model generates 16 rollouts. The initial tool invocation reward coefficient $\lambda_{\text{tool}}^{\text{base}}$ is set to 1.2, and the clipping parameter $\epsilon$ is set to 0.2. We adopt the AdamW optimizer with an initial learning rate of $1 \times 10^{-6}$. The maximum response length of the policy model is set to 20,480 tokens.

### 4.3. Main Results

**Perception benchmarks.** Table 1 presents the comparison results of our AutoTool with existing models on perception benchmarks. All models first generate internal reasoning before producing a final answer. Visual grounding reasoning models rely on their respective system prompts to trigger tool usage, whereas AutoTool adaptively decides whether to invoke tools using the same system prompt as during training. Across the majority of splits in both datasets, AutoTool consistently achieves state-of-the-art performance, significantly surpassing both proprietary and open-source general MLLMs. Remarkably, AutoTool still maintains a clear advantage even over much larger models such as Qwen2.5-VL-32B and InternVL3-38B. Compared with models that also rely on visual grounding–based reasoning, our approach breaks away from a single reasoning paradigm, effectively leveraging the advantages of localized reasoning after accurate grounding, while avoiding redundant or misleading information introduced by unnecessary or incorrect localization. Compared with the base model Qwen2.5-VL-7B, our training paradigm leads to substantial improvements on perception tasks, achieving 21% and 11% accuracy gains on HRbench and V* datasets, respectively.

**Grounding benchmarks.** As shown in Table 2, AutoTool consistently outperforms models of comparable size across all splits of the four datasets. This improvement stems from our training design: in the `<tool_on>` mode, trajectories that correctly invoke tools and produce accurate answers are rewarded, which encourages the model to precisely localize

*Table 1.* Performance comparison of different models on perception benchmarks. For models with similar sizes, the best performance for each metric is marked as **bold**, and the second-best is underlined.

| Model | Size | HRbench-4K | | | HRbench-8K | | | V* | | |
| --- | --- | --- | --- | --- | --- | --- | --- | --- | --- | --- |
| | | FSP | FCP | Overall | FSP | FCP | Overall | Attribute | Spatial | Overall |
| ***Proprietary General MLLMs*** | | | | | | | | | | |
| GPT-4o [2023] (Achiam et al., 2023) | - | 66.8 | 63.3 | 65.0 | 60.8 | 58.5 | 59.6 | 72.2 | 60.5 | 67.5 |
| o3 [2025] (OpenAI, 2025) | - | - | - | - | - | - | - | - | - | 95.7 |
| ***Open-Source General MLLMs*** | | | | | | | | | | |
| Qwen2.5-VL-7B [2025] (Bai et al., 2025) | 7B | 81.8 | 57.5 | 69.6 | 74.0 | 52.0 | 63.0 | 67.0 | 72.4 | 69.1 |
| Qwen2.5-VL-32B [2025] (Bai et al., 2025) | 32B | 87.5 | 65.0 | 76.3 | 85.8 | 61.8 | 73.8 | 78.3 | 80.3 | 79.1 |
| InternVL3-8B [2025] (Zhu et al., 2025) | 8B | 75.8 | **63.8** | 69.8 | 59.0 | 58.8 | 58.9 | 75.7 | 81.6 | 78.0 |
| InternVL3-38B [2025] (Zhu et al., 2025) | 38B | 81.0 | 67.0 | 74.0 | 64.8 | 60.3 | 62.5 | 78.3 | 77.6 | 78.0 |
| LLaVA-OneVision [2024] (Li et al., 2024a) | 7B | 72.8 | 53.8 | 63.3 | 60.5 | 50.3 | 55.4 | 74.8 | 68.4 | 72.3 |
| ***Visual Grounded Reasoning Models*** | | | | | | | | | | |
| SEAL [2024] (Wu & Xie, 2024) | 7B | - | - | - | - | - | - | 74.8 | 76.3 | 75.4 |
| DyFo [2025] (Li et al., 2025) | 7B | - | - | - | - | - | - | 80.0 | 82.9 | 81.2 |
| ZoomEye [2025] (Shen et al., 2024) | 7B | 84.3 | 55.0 | 69.6 | **88.5** | 50.0 | 69.3 | **93.9** | 85.5 | **90.6** |
| Pixel-Reasoner [2025] (Su et al., 2025a) | 7B | 86.0 | 60.3 | 72.9 | 80.0 | 54.4 | 66.9 | 83.5 | 76.3 | 80.6 |
| DeepEyes [2025] (Zheng et al., 2025) | 7B | 92.0 | 57.8 | 74.9 | 85.5 | 57.5 | 71.5 | 90.4 | 82.9 | 87.4 |
| ***Ours*** | | | | | | | | | | |
| AutoTool | 7B | **92.5** | 61.3 | **76.9** | 88.0 | **60.0** | **74.0** | 91.3 | **88.2** | 90.1 |
| $\Delta v.s.$ Qwen2.5-VL-7B | - | ↑ 10.7 | ↑ 3.8 | ↑ 7.3 | ↑ 14.0 | ↑ 8.0 | ↑ 11.0 | ↑ 24.3 | ↑ 15.8 | ↑ 21.0 |

*Table 2.* Performance comparison of different models on grounding benchmarks. The best performance for each metric is marked as **bold**.

| Model | Size | refCOCO | | | | refCOCOg | | refCOCO+ | | | ReasonSeg | |
| --- | --- | --- | --- | --- | --- | --- | --- | --- | --- | --- | --- | --- |
| | | test | testA | testB | val | test | val | testA | testB | val | test | val |
| Qwen2.5-VL-7B (Bai et al., 2025) | 7B | 84.7 | 86.6 | 78.1 | 83.4 | 77.0 | 76.6 | 82.1 | 68.5 | 76.3 | 51.1 | 59.5 |
| DeepEyes (Zheng et al., 2025) | 7B | 86.0 | 90.5 | 79.6 | 86.1 | 80.3 | 80.4 | 87.2 | 67.8 | 79.2 | 50.6 | 61.5 |
| AutoTool | 7B | **88.5** | **92.5** | **83.1** | **88.6** | **82.8** | **82.7** | **89.7** | **72.6** | **81.6** | **53.3** | **63.0** |
| $\Delta v.s.$ Qwen2.5-VL-7B | - | ↑ 3.8 | ↑ 5.9 | ↑ 5.0 | ↑ 5.2 | ↑ 5.8 | ↑ 6.1 | ↑ 7.6 | ↑ 4.1 | ↑ 5.3 | ↑ 2.2 | ↑ 3.5 |

the region of interest. Conversely, trajectories that invoke tools but yield incorrect answers are penalized, reducing the likelihood of erroneous localizations. In contrast, DeepEyes may still rewards trajectories where tool-based localization is incorrect but the final answer happens to be correct. Our introduction of the `<tool_off>` mode mitigates this issue by encouraging reasoning without relying on potentially misleading tool-based cues. All models are evaluated using the same prompt, and the detailed prompt specifications are provided in the supplementary material.

**Hallucination and reasoning benchmark.** As illustrated in Table 3, our model demonstrates improved performance in reducing hallucinations. The adaptive tool invocation capability is also effective for hallucination tasks: when determining whether a target object is present in the image, AutoTool carefully inspects similar objects or potential regions in the `<tool_on>` mode. Consistent with the perception benchmarks, all models first generate internal reasoning before providing a final answer, with DeepEyes and AutoTool leveraging tool invocation. Our model maintains robust reasoning capabilities and achieves excellent performance across six benchmarks encompassing a diverse

range of reasoning tasks. All models are evaluated under the same prompt setting, where each model first conducts internal reasoning before producing the final answer.

We further showcase visual examples on the test benchmarks, as detailed in the supplementary material.

### 4.4. Ablation and Analysis

**The influence of each module.** As shown in Table 4, **Tool on** denotes the reasoning process assisted by the zoom-in tool, while **Tool off** represents pure text-based reasoning. **MSPO_penalty** refers to the negative reward $R_{\text{tool}} = -0.5$ applied when the model invokes a tool but produces an incorrect answer. **AMB_free** indicates that the AMB constraint is removed in the later training stage, allowing the model to freely explore dual reasoning modes.

In the #1 setting, Qwen2.5-VL-7B is trained on the training dataset via pure-text GRPO, which substantially improves performance over the base model. In #2, DeepEyes always employs the zoom-in tool for every query, leading to further improvement compared with #1 setting. In #3, a carefully designed prompt with RL training is adopted under mode-

*Table 3.* Performance comparison on hallucination and reasoning benchmarks. The best performance for each metric is marked as **bold**.

| Model | Size | POPE | | | | MathVista | MathVerse | MathVision | | WeMath | DynaMath | LogicVista |
|---|---|---|---|---|---|---|---|---|---|---|---|---|
| | | Adversarial | Popular | Random | Overall | | | test | testmini | | | |
| Qwen2.5-VL-7B (Bai et al., 2025) | 7B | 85.9 | 87.0 | 88.9 | 87.2 | 70.6 | 43.6 | 14.8 | 16.1 | 30.8 | 57.2 | 45.5 |
| InternVL3-8B (Zhu et al., 2025) | 8B | **87.2** | 88.1 | 90.8 | 88.7 | 68.3 | **47.8** | **17.4** | 16.8 | **37.9** | 57.8 | 46.0 |
| LLaVA-OneVision (Li et al., 2024a) | 7B | 84.7 | 87.6 | 90.4 | 87.6 | 58.4 | 34.5 | 9.6 | 12.5 | 37.5 | 35.4 | 28.3 |
| DeepEyes (Zheng et al., 2025) | 7B | 81.4 | 85.0 | 91.7 | 86.0 | 71.6 | 45.2 | 15.1 | 19.4 | 32.6 | 57.7 | 45.3 |
| AutoTool | 7B | 86.1 | **88.4** | **92.3** | **88.9** | **72.8** | 45.9 | 15.0 | **19.4** | 34.0 | **58.0** | **46.7** |
| $\Delta v.s.$ Qwen2.5-VL-7B | - | ↑ 0.2 | ↑ 1.4 | ↑ 3.4 | ↑ 1.7 | ↑ 2.2 | ↑ 2.3 | ↑ 0.2 | ↑ 3.3 | ↑ 3.2 | ↑ 0.8 | ↑ 1.2 |

*Table 4.* Ablation experiments of each module. The best performance for each metric is marked as **bold**.

| ID | Tool on | Tool off | MSPO$_{penalty}$ | AMB$_{free}$ | HRbench-4K | | | HRbench-8K | | | V* | | |
|---|---|---|---|---|---|---|---|---|---|---|---|---|---|
| | | | | | FSP | FCP | Overall | FSP | FCP | Overall | Attribute | Spatial | Overall |
| 1 | | ✓ | | | 88.0 | 59.3 | 73.6 | 81.8 | 58.8 | 70.2 | 87.8 | 81.6 | 85.3 |
| 2 | ✓ | | | | 92.0 | 57.8 | 74.9 | 85.5 | 57.5 | 71.5 | 90.4 | 82.9 | 87.4 |
| 3 | ✓ | ✓ | | | 91.8 | 58.8 | 75.3 | 86.3 | 58.5 | 72.4 | 88.7 | 88.2 | 88.5 |
| 4 | ✓ | ✓ | ✓ | | **93.3** | 58.3 | 75.8 | 87.3 | 59.3 | 73.3 | 89.6 | 88.2 | 89.0 |
| 5 | ✓ | ✓ | | ✓ | 92.8 | 60.8 | 76.8 | 85.0 | **61.5** | 73.2 | 89.6 | **89.5** | 89.5 |
| 6 | ✓ | ✓ | ✓ | ✓ | 92.5 | **61.3** | **76.9** | **88.0** | 60.0 | **74.0** | **91.3** | 88.2 | **90.1** |

*Table 5.* Effect of removing the mode-balancing constraint at different training steps. The best performance is marked as **bold**.

| Step | HRbench-4K | | | HRbench-8K | | | V* | | |
|---|---|---|---|---|---|---|---|---|---|
| | FSP | FCP | Overall | FSP | FCP | Overall | Attribute | Spatial | Overall |
| 0 | 92.5 | 55.8 | 74.1 | 84.5 | 57.3 | 70.9 | 85.3 | 84.2 | 84.3 |
| 50 | 92.8 | 60.8 | 76.8 | 85.0 | **61.5** | 73.2 | 90.4 | 86.8 | 89.0 |
| 60 | 92.5 | **61.3** | **76.9** | 88.0 | 60.0 | **74.0** | **91.3** | 88.2 | **90.1** |
| 70 | 92.5 | 59.5 | 76.0 | **89.0** | 58.0 | 73.5 | 89.6 | **89.5** | 89.5 |
| 80 | **93.3** | 58.3 | 75.8 | 87.3 | 59.3 | 73.3 | 89.6 | 88.2 | 89.0 |

*Table 6.* Effect of the initial tool invocation reward coefficient $\lambda_{tool}^{base}$. The best performance is marked as **bold**.

| $\lambda_{tool}^{base}$ | 0.0 | 0.5 | 1.0 | 1.2 | 1.4 | 3.0 | 5.0 |
|---|---|---|---|---|---|---|---|
| HRbench-4K | 72.4 | 75.5 | 76.0 | 76.9 | **77.3** | 75.0 | 71.4 |
| HRbench-8K | 69.8 | 72.8 | 73.9 | **74.0** | 73.5 | 72.1 | 68.4 |
| V* | 84.3 | 88.5 | 89.5 | **90.1** | **90.1** | 88.0 | 83.8 |

*Table 7.* Sensitivity analysis of the efficiency penalty term.

| Penalty | HRbench-4K | | | HRbench-8K | | | V* | | |
|---|---|---|---|---|---|---|---|---|---|
| | FSP | FCP | Overall | FSP | FCP | Overall | Attribute | Spatial | Overall |
| 0 | 92.8 | 60.8 | 76.8 | 85.0 | 61.5 | 73.2 | 89.6 | 89.5 | 89.5 |
| -0.2 | 92.3 | 61.0 | 76.6 | 87.5 | 59.5 | 73.5 | 91.3 | 86.8 | 89.5 |
| -0.5 | 92.5 | 61.3 | 76.9 | 88.0 | 60.0 | 74.0 | 91.3 | 88.2 | 90.1 |
| -0.8 | 92.5 | 61.5 | 77.0 | 87.8 | 59.5 | 73.6 | 90.4 | 89.5 | 90.1 |

*Table 8.* Training and inference efficiency comparison between AutoTool and DeepEyes.

| Phase | Dataset | Split | DeepEyes | AutoTool | Speedup (%) |
|---|---|---|---|---|---|
| **Training** | - | - | 44.9 h | 35.8 h | 20.3 |
| **Inference** | **V*** | Direct | 2.23 min | 1.68 min | 24.7 |
| | | Relative | 1.50 min | 1.07 min | 28.7 |
| | **HRbench** | 4K | 48.35 min | 31.95 min | 33.9 |
| | | 8K | 53.45 min | 33.08 min | 38.1 |
| | **POPE** | Adversarial | 14.27 min | 8.78 min | 38.5 |
| | | Popular | 12.13 min | 7.73 min | 36.3 |
| | | Random | 13.07 min | 7.20 min | 44.9 |

ratio constraints, guiding the model to fully explore both modes. Compared with #2 setting, this flexible reasoning mode mitigates the negative impact of incorrect tool usage. In the #4 setting, a penalty is introduced when the model invokes a tool but produces an incorrect answer, enforcing more precise grounding behavior. In #5, the mode-balancing constraint is removed in the later training stage encouraging free exploration and yielding further performance gains. Finally, the #6 setting integrates all these advantageous components and achieves the best overall performance.

**Effect of mode-balancing removal step.** Table 5 reports the impact of removing the mode-balancing constraint at different training steps. We observe that disabling the AMB constraint from the beginning (*i.e.*, step 0) leads to a premature dominance of the `<tool_off>` mode, resulting in inferior performance on fine-grained perception tasks. As training progresses, the model benefits from maintaining the constraint for a sufficient period, which promotes balanced exploration between the two reasoning modes. The best

overall results are achieved for HRBench-4K, HRBench-8K, and V* benchmarks when the constraint is removed at around 60 iterations. Further delaying the removal (*e.g.*, step 70) yields a slight performance decline, likely because the model becomes overly constrained and less adaptive to problem-specific reasoning strategies in later stages.

**Effect of the coefficient $\lambda_{tool}^{base}$.** We further analyze the sensitivity of the initial tool invocation reward coefficient $\lambda_{tool}^{base}$, including both moderate and extreme settings. The results are shown in Table 6. The model achieves stable performance around the default value ($\lambda_{tool}^{base} = 1.2$) and remains robust within a moderate range ($0.5 \sim 3.0$), suggesting that AMB is not sensitive to precise hyperparameter tuning.

However, extreme values lead to clear performance degradation due to reward imbalance. When $\lambda_{tool}^{base}$ is too small, the contribution of the tool reward becomes negligible, weakening supervision on tool-usage behavior and causing re-

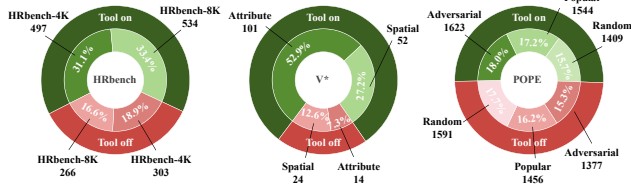

*Figure 3.* The outer ring shows the proportion of the dual reasoning modes on three datasets, while the inner ring presents their distribution across different splits within each dataset.

ward hacking. For example, when $\lambda_{\text{tool}}^{\text{base}} = 0.0$, the model frequently selects the `<tool_on>` mode while performing pure text reasoning without valid tool calls, exploiting $R_{\text{tool}}^{\text{on}} = 0$ while benefiting from a larger $\lambda_{\text{tool}}^{\text{off}}$. In contrast, excessively large values reduce the relative influence of the task reward and bias the policy toward the foundation model preference, resulting in collapse to the `<tool_off>` mode. We additionally observe degraded adherence to the required reasoning format under such settings. Overall, the default coefficient provides a good balance between adaptive tool usage and reasoning quality.

**Sensitivity to the efficiency penalty.** The efficiency penalty in the mode-specific reward is introduced to discourage unnecessary tool usage, particularly when tool invocation leads to incorrect answers. Specifically, for the `<tool_on>` mode, trajectories with incorrect answers after tool invocation receive a negative reward. To evaluate the sensitivity of this design, we vary the efficiency penalty term within $\{0, -0.2, -0.5, -0.8\}$ while keeping all other hyperparameters unchanged. The results are shown in Table 7. Empirically, the performance varies only marginally across different penalty values, indicating that the proposed reward design maintains a robust operating range and does not require careful dataset-specific tuning.

**Time efficiency and tool mode analysis.** As shown in Table 8, we report the training and inference time costs of existing visual grounding–based reasoning models such as DeepEyes and our AutoTool. Under the same data and number of training iterations, our method reduces the total training time by approximately 9 hours. The inference time across all three datasets is also significantly shortened, with a 44.9% reduction observed on the POPE dataset.

In addition, we analyze the occurrence frequency of the dual reasoning modes across different benchmarks. As illustrated in Figure 3, the proportion of these two modes is not fixed as in the training stage but rather dynamically varies depending on the characteristics of the dataset. For high-resolution datasets such as HRbench and V*, where target objects often occupy a small region of the image, the `<tool_on>` mode appears more frequently. In contrast, POPE contains relatively smaller images with larger target objects, leading to a notably higher proportion of `<tool_off>` mode during inference. The ratio of the dual reasoning modes during

training is illustrated in Figure 1(c). In the early and middle stages of training, we adaptively control the reward factor to encourage sufficient exploration of both reasoning modes, resulting in a roughly balanced distribution of about 50% for each. In the later stage, we remove this constraint to allow the model to freely choose its preferred reasoning strategy, where a slight increase in the proportion of the `<tool_on>` mode can be observed.

## 5. Conclusion

In this work, we reveal that tool-augmented reasoning is not always beneficial for MLLMs. To address this limitation, we propose AutoTool, a model that dynamically determines whether to invoke zoom-in tools based on the characteristics of each query. This design significantly improves both training and inference efficiency while mitigating the adverse effects of unnecessary or incorrect tool usage. Based on the reinforcement learning framework, our approach optimizes dual reasoning modes with carefully designed reward functions and guides the model to fully explore both. Extensive experiments on various benchmarks demonstrate that AutoTool achieves superior reasoning capability and efficiency compared to existing models.

## Acknowledgements

This work was supported by NSFC Project (62536005, 62192783, 62506162), Jiangsu Science and Technology Project (BG2024031, BK20251241), Fundamental and Interdisciplinary Disciplines Breakthrough Plan of the Ministry of Education of China (No. JYB2025XDXM118), "111 Center" (No. B26023), and Fundamental Research Funds for the Central Universities (KG202508).

## Impact Statement

This work studies adaptive tool invocation for multimodal large language models (MLLMs). By enabling models to selectively determine whether external tool assistance is necessary, our method improves reasoning efficiency while reducing redundant computation and unnecessary tool interactions. Our approach also reduces hallucinations caused by inappropriate tool usage, potentially improving the reliability of multimodal reasoning systems. However, the proposed method does not eliminate risks associated with MLLMs, such as incorrect reasoning or failures in complex scenarios. Careful evaluation and appropriate human oversight remain important for real-world deployment.

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

# A. Prompt Design for Training and Evaluation

To ensure consistent interaction between the model and the environment, we carefully design the system and user prompts for both the training and inference stages. The prompts aim to explicitly distinguish between the dual reasoning modes, guiding the model to generate reasoning trajectories accordingly.

## A.1. Training Prompts

During training, the system prompt defines the model's role and available reasoning modes. The user prompt provides multimodal inputs (text and images) and instructs the model to reason under the specified mode.

**System Prompt:**

> You are a helpful assistant.
> At the beginning of your first response, you must output either <tool_on>or <tool_off>to indicate whether tools will be used to assist with subsequent answers.
>   - <tool_on>means that you may call tools to help answer the query.
>   - <tool_off>means that you will answer entirely without tool usage.
>
> # When to choose <tool_on>
> Use <tool_on>if the question requires close inspection or verification of fine details in an image, such as:
>   - identifying a specific object among multiple objects,
>   - checking small or unclear regions, sub-tables, or fine textures,
>   - verifying visual details that may affect the correctness of the answer.
> In these cases, call the zoom-in tool as needed to focus on the relevant region.
>
> # When to choose <tool_off>
> Use <tool_off>if:
>   - the question needs global or overall image understanding (scene, layout, general context), or the relevant region or object is already clear enough without zooming in,
>   - zooming in would not provide useful additional information.
>
> # Tool calling format
> You may call one or more functions to assist with the user query.
> You are provided with function signatures within <tools></tools>XML tags:
> <tools>
> {"type": "function", "function": {"name": "image_zoom_in_tool", "description": "Zoom in on a specific region of an image by cropping it based on a bounding box (bbox) and an optional object label.", "parameters": {"type": "object", "properties": {"bbox_2d": {"type": "array", "items": {"type": "number"}, "minItems": 4, "maxItems": 4, "description": "The bounding box of the region to zoom in, as [x1, y1, x2, y2], where (x1, y1) is the top-left corner and (x2, y2) is the bottom-right corner."}, "label": {"type": "string", "description": "The name or label of the object in the specified bounding box (optional)."}}, "required": ["bbo_2d"]}}}
> </tools>
>
> # How to call a tool
> Return a json object with function name and arguments within <tool_call></tool_call>XML tags:
> <tool_call>
> {"name": <function-name>, "arguments": <args-json-object>}
> </tool_call>
>
> **Example**:
> <tool_call>
> {"name": "image_zoom_in_tool", "arguments": {"bbox_2d": [10, 20, 100, 200], "label": "the apple on the desk"}}
> <tool_call>

**User Prompt:**

Question:{question}

Please follow these instructions strictly:
1. First, determine whether you will use a tool by outputting <tool_on>or <tool_off>.
2. Then, show your reasoning inside <think>... </think>.
3. If tool usage is required (<tool_on>), call the image_zoom_in_tool using <tool_call>...</tool_call>, and DO NOT provide an <answer>yet — wait for the zoomed image in the next round.
4. If no tool is needed (<tool_off>), provide your final answer inside <answer>... </answer>.

Format strictly as:
<tool_on>  <think>  ...  </think>  <tool_call>  ...  </tool_call>  OR  <tool_off>  <think>  ... </think>  <answer>  ...  </answer>

## A.2. Evaluation Prompts

For the Perception and Hallucination benchmarks, we use the same prompt as in the training phase to evaluate the model's ability in adaptive tool invocation. For the Reasoning datasets, we adopt the official prompts provided by each benchmark. For the Grounding benchmark, following Seg-Zero (Liu et al., 2025), we employ the user prompt template as:

**System Prompt:**

You are a helpful assistant.

**User Prompt:**

Please find "{Question}" with bboxs.
Compare the difference between object(s) and find the most closely matched object(s).
Output the thinking process in <think></think>and final answer in <answer></answer>tags.
Output the bbox(es) inside the interested object(s) in JSON format.
i.e. <think>thinking process here </think>   <answer>{{"bbox_2d": [10,100,200,210] } }</answer>

# B. Reward Function Details

**Accuracy reward** $R_{\text{acc}}$**:** We evaluate whether the predicted answer is semantically equivalent to the ground truth. The reward takes values in $\{0, 0.8\}$, where 0 corresponds to an incorrect answer and $0.8$ to a correct one. Evaluation is performed using a combination of rule-based metrics and an online reward model (*e.g.* Qwen2.5-72B-Instruct). Specifically, we first perform an exact string matching between the model output and the ground-truth answer. If the two strings are identical, the prediction is directly regarded as correct. Otherwise, we further evaluate semantic equivalence via an online reward model. The reward model is prompted to judge whether the predicted answer conveys the same meaning as the ground truth under a fixed system prompt, as detailed below.

**User Prompt:**

Below are two answers to a question. Question is [Question], [Standard Answer] is the standard answer to the question, and [Model_answer] is the answer extracted from a model's output to this question. Determine whether these two answers are consistent.

Note that [Model Answer] is consistent with [Standard Answer] whenever they are essentially the same. If the meaning is expressed in the same way, it is considered consistent, for example, 'pink' and 'it is pink'.

If they are consistent, Judement is 1; if they are different, Judement is 0. Just output Judement and don't output anything else.

[Question]: Is the countertop tan or blue?
[Standard Answer]: The countertop is tan.
[Model_answer] : tan
Judgement: 1

[Question]: On which side of the picture is the barrier?
[Standard Answer]: The barrier is on the left side of the picture.
[Model_answer] : left
Judgement: 1

[Question]: Is the kite brown and large?
[Standard Answer]: Yes, the kite is brown and large.
[Model_answer] : Yes
Judgement: 1

[Question]: Are the spots on a giraffe?
[Standard Answer]: No, the spots are on a banana.
[Model_answer] : no
Judgement: 1

[Question]: Who is wearing pants?
[Standard Answer]: The boy is wearing pants.
[Model_answer] : The person in the picture is wearing pants.
Judgement: 1

[Question]: Is the man phone both blue and closed?
[Standard Answer]: Yes, the man phone is both blue and closed.
[Model_answer] : No.
Judgement: 0

[Question]: What color is the towel in the center of the picture?
[Standard Answer]: The towel in the center of the picture is blue.
[Model_answer] : The towel in the center of the picture is pink.
Judgement: 0

[Question]: {question}
[Standard Answer]: {ground_truth}
[Model_answer] : {predict_str}
Judgement:

As shown above, the prompt provides seven illustrative examples covering both consistent and inconsistent cases. The target question is placed at the end. The reward model is instructed to output a binary judgment, where `Judgement = 1` indicates semantic consistency between the prediction and the standard answer, and `Judgement = 0` otherwise. This hybrid evaluation strategy combines strict rule-based verification with flexible semantic evaluation, enabling reliable supervision for both factual and open-ended responses.

**Format reward** $R_{\textbf{format}}$**:** This reward ensures that the reasoning process and final answer adhere to the prescribed output format, i.e., enclosed within `<think></think>` and `<answer></answer>` tags. The reward takes values in $\{-0.2, 0\}$, where $-0.2$ indicates a format violation and $0$ indicates correct formatting.

**Mode-specific tool reward** $R_{\text{tool}}$**:** The computation of $R_{\text{tool}}$ follows the procedure described in Section 3.4. The reward is further modulated by $\lambda_{\text{tool}}^{\text{mode}}$ as defined in Equation (5).

## C. Comparison with Prior Tool-Use Methods in LLMs

Prior work on adaptive tool use in LLMs, such as ReAct (Yao et al., 2022) and Toolformer (Schick et al., 2023), primarily relies on prompt structures or local training signals to guide tool invocation. ReAct interleaves reasoning and actions through carefully designed prompts, enabling the model to decide whether and how to call tools during generation. Toolformer introduces a self-supervised objective that retains tool calls based on changes in prediction cross-entropy with and without tool usage.

While effective, these approaches determine tool usage based on local or proxy signals, rather than directly assessing whether tool invocation is necessary for producing a correct final answer. In contrast, our method rolls out complete reasoning trajectories under different reasoning modes (*e.g.* `<tool_on>` and `<tool_off>`) and rewards trajectories based on answer correctness. This outcome-driven formulation allows AutoTool to learn when tool invocation is genuinely beneficial, without relying on intermediate heuristics.

## D. Detailed Training Information

To gain a deeper understanding of how AutoTool learns to balance and adapt its reasoning behaviors, we further analyze training dynamics, focusing on the distribution of reasoning modes, the trends in tool invocation frequency and response length as shown in Figure 4.

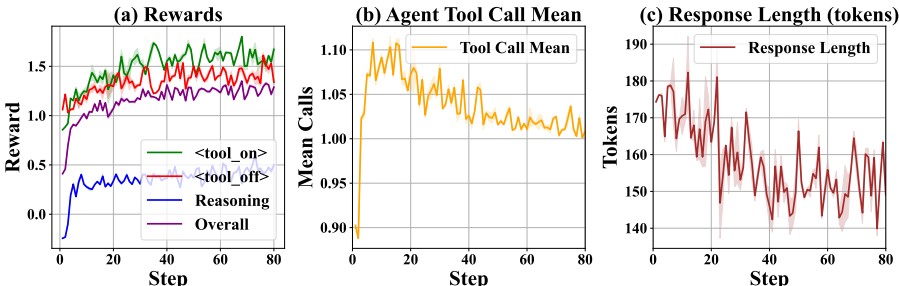

*Figure 4.* Detailed training-phase analysis. (a) Dual reasoning trajectories, general reasoning data, and overall average reward curves. (b) Average number of tool invocations under the `<tool_on>` mode. (c) Response length variations throughout training. The shaded regions denote the standard deviation across multiple runs.

For each batch of training data, we add tool-invocation prompts to the samples from the V* (where the V* here is distinct from the V* used in evaluation benchmarks) and ArxivQA datasets, while the samples from the ThinkLite-VL dataset adopt purely text-centric reasoning and answer generation to preserve general reasoning capability. Figure 4(a) presents the average reward curves for three reasoning types, along with the overall average reward for all samples. All rewards show a steady upward trend, demonstrating the effectiveness of our training strategy. Figure 4(b) illustrates the average number of tool invocations in `<tool_on>` reasoning trajectories during training. The model quickly learns the correct invocation format in the early stage, and the average number of tool calls gradually stabilizes just above one per query, reflecting a more deliberate and efficient tool-usage behavior. Figure 4(c) shows the curve of the average number of generated tokens, which gradually decreases and stabilizes around 150. Combined with the increasing reward trend, this indicates that our method enables the model to produce more accurate answers with lower reasoning cost.

## E. Accuracy Analysis under Different Reasoning Modes

We report the accuracy under different reasoning modes across three benchmarks in Figure 5. As shown, the error rate of the `<tool_off>` mode is consistently lower than that of the tool-on mode on all benchmarks (10.8% vs. 13.8%, 1.6% vs. 8.4%, and 4.4% vs. 6.7%, respectively).

This observation is expected, as AutoTool tends to select the `<tool_off>` mode for relatively simple queries that can be reliably solved using the model's internal knowledge alone. In such cases, invoking external tools may introduce unnecessary

operations or error propagation, leading to higher failure rates. In contrast, the `<tool_on>` mode is predominantly activated for more complex or visually challenging questions, where the overall task difficulty is inherently higher.

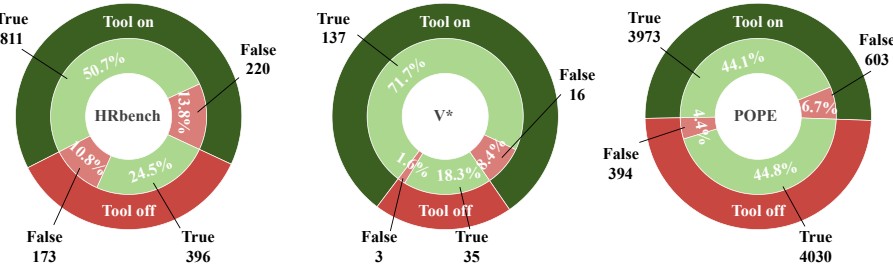

*Figure 5.* Accuracy comparison between both reasoning modes across three benchmarks.

# F. Comparison with other Baselines

We further compare AutoTool with prompt-based baselines on Qwen2.5-VL-7B (Bai et al., 2025), where reasoning modes are controlled solely through prompt design. We additionally include LlamaV-o1 (Thawakar et al., 2025) for comparison, which is built upon Llama-3.2-11B-Vision-Instruct (Meta, 2024) and trained via SFT to follow a fixed reasoning pattern, generating detailed reasoning steps and final answers after producing summaries and descriptions. As shown in Table 9, although prompt engineering can partially affect tool usage, it lacks reliable and stable control. In contrast, our RL-based method (MSPO+AMB) consistently outperforms prompt-only baselines. Notably, AutoTool outperforms LlamaV-o1 even with a smaller model size, highlighting the efficiency and adaptability of RL-based strategy.

*Table 9.* Comparison with other Baselines.

| Exp | HRbench-4K | | HRbench-8K | | V* | | POPE | | |
|---|---|---|---|---|---|---|---|---|---|
| | FSP | FCP | FSP | FCP | Attribute | Spatial | Adversarial | Popular | Random |
| Base Model_prompt | 87.8 | 60.3 | 82.5 | 58.5 | 87.0 | 81.6 | 80.1 | 80.3 | 80.8 |
| LlamaV-o1 | 66.5 | 56.3 | 52.3 | 49.0 | 60.0 | 79.0 | 85.1 | 87.8 | 90.4 |
| AutoTool | **92.5** | **61.3** | **88.0** | **60.0** | **91.3** | **88.2** | **86.1** | **88.4** | **92.3** |

# G. Results on Other Base Models

To evaluate the robustness of our method across different base models, we conduct additional experiments using Qwen2.5-VL-3B as the foundation model. Training is performed on four H200 GPUs, while an additional two H200 GPUs are used to deploy the reward model. Table 10 reports results on a diverse set of benchmarks. Across all benchmarks, AutoTool_3B consistently outperforms the corresponding base model. These results indicate that the proposed method is not tied to a specific model scale and generalizes well to other base models.

*Table 10.* Results on Other Base Models

| Exp | HRbench-4K | | HRbench-8K | | V* | | POPE | | |
|---|---|---|---|---|---|---|---|---|---|
| | FSP | FCP | FSP | FCP | Attribute | Spatial | Adversarial | Popular | Random |
| Base Model-3B | 87.8 | 60.3 | 82.5 | 58.5 | 87.0 | 81.6 | 80.1 | 80.3 | 80.8 |
| AutoTool-3B | **92.5** | **61.3** | **88.0** | **60.0** | **91.3** | **88.2** | **86.1** | **88.4** | **92.3** |

| Exp | refCOCO | | | | refCOCOg | | refCOCO+ | | |
|---|---|---|---|---|---|---|---|---|---|
| | test | testA | testB | val | test | val | testA | testB | val |
| Base Model-3B | 82.0 | 46.0 | 77.3 | 42.8 | 52.2 | 56.6 | 73.1 | 73.5 | 74.0 |
| AutoTool-3B | **86.5** | **54.0** | **81.8** | **49.3** | **79.1** | **61.8** | **85.6** | **87.4** | **91.77** |

| Exp | ReasonSeg | | MathVista | MathVerse | MathVision | | WeMath | DynaMath | LogicVista |
|---|---|---|---|---|---|---|---|---|---|
| | test | val | | | test | testmini | | | |
| Base Model-3B | 28.4 | 34.0 | 56.5 | 33.2 | 11.7 | 14.1 | 23.1 | 47.0 | 40.6 |
| AutoTool-3B | **41.9** | **52.0** | **62.5** | **36.0** | **12.8** | **17.4** | **33.5** | **50.1** | **41.7** |

## H. Additional Comparison on Hallucination Benchmarks

HallusionBench is a benchmark that evaluates both visual illusion and knowledge hallucination in MLLMs. We conduct experiments on its image split to complement POPE, which mainly focuses on object existence hallucination. For all baseline models, including Qwen2.5-VL-7B (Bai et al., 2025), InternVL3-8B (Zhu et al., 2025), LLaVA-OneVision-7B (Li et al., 2024a), and DeepEyes-7B (Zheng et al., 2025), we use the same prompting strategy as in the POPE experiments. As shown in Table 11, AutoTool achieves the best performance on HallusionBench, demonstrating that our method generalizes beyond POPE and effectively reduces both visual and knowledge hallucinations.

*Table 11.* Comparison of different models on HallusionBench.

| Benchmark | Qwen2.5-VL-7B | InternVL3-8B | LLaVA-OneVision-7B | DeepEyes-7B | AutoTool-7B |
|---|---|---|---|---|---|
| HallusionBench | 57.8 | 59.7 | 52.9 | 58.5 | **60.9** |

## I. Further Ablation Studies

To further analyze the design choices of our method, as shown in Table 12, we conducted additional ablation experiments. AutoTool$_{\text{SFT}}$ performs an additional SFT stage before GRPO using a small amount of data that matches the dual reasoning mode. Although the model learns both reasoning types, this rigid and forced training procedure disrupts the model's inherent knowledge, leading to a significant performance drop.

AMB$_{\text{linear}}$ linearly decreases the influence of $F_{on}$ on $\lambda_{\text{tool}}^{\text{mode}}$ during training, following $\lambda_{\text{tool}}^{\text{mode}} = \lambda_{\text{tool}} \pm \frac{t}{t_{\max}}(0.5 - F_{\text{on}})$, where $t$ denotes the current training step and $t_{\max}$ represents the total number of training steps. This schemes still impose a residual constraint throughout training, merely reducing its strength over time without granting the model full freedom. AutoTool$_{\text{w/o AMB}}$ conduct an ablation without the AMB module, leaving the rollout proportions of the two reasoning modes uncontrolled. Due to the inherent reasoning bias of the foundation model, the policy strongly favors <tool_off>, converging to pure text-based reasoning. Results show a clear performance drop compared to AutoTool, highlighting the importance of balanced mode constraint. In contrast, our method focuses on balanced exploration of dual reasoning modes during the early and middle stages of training, and completely removes the constraint in the later stage, allowing the model to freely explore and consolidate its preferred reasoning strategy.

*Table 12.* Further Ablation experiments.

| Exp | HRbench-4K | | | HRbench-8K | | | V* | | |
|---|---|---|---|---|---|---|---|---|---|
| | FSP | FCP | Overall | FSP | FCP | Overall | Attribute | Spatial | Overall |
| AutoTool$_{\text{SFT}}$ | 59.0 | 56.5 | 57.8 | 52.0 | 53.3 | 52.6 | 50.4 | 64.5 | 56.0 |
| AMB$_{\text{linear}}$ | 92.3 | 59.3 | 75.8 | 86.5 | 58.5 | 72.5 | 90.4 | 86.8 | 89.0 |
| AutoTool$_{\text{w/o AMB}}$ | 87.8 | 60.3 | 74.0 | 82.5 | 58.5 | 70.5 | 87.0 | 81.6 | 84.8 |
| AutoTool$_{\text{delay}}$ | **92.7** | **61.3** | **77.0** | 87.5 | **60.3** | 73.9 | 90.4 | 88.2 | 89.5 |
| AutoTool$_{\text{notoken}}$ | 92.3 | 60.8 | 76.5 | 87.0 | 60.0 | 73.5 | 89.6 | **89.5** | 89.5 |
| AutoTool$_{\text{w/ KL}}$ | 91.5 | 59.8 | 75.6 | 86.5 | 59.3 | 72.9 | 89.6 | 88.2 | 89.0 |
| AutoTool | 92.5 | **61.3** | 76.9 | **88.0** | 60.0 | **74.0** | **91.3** | 88.2 | **90.1** |

To study whether the first-step tool decision is overly restrictive, we also evaluate a variant that delays the generation of <tool_on> or <tool_off> until after an explicit thinking phase. Specifically, the model follows <think> ... </think> <tool_on> <tool_call> ... </tool_call> or <think> ... </think> <tool_off> <answer> ... </answer>. The results are reported as AutoTool$_{\text{delay}}$. We observe no significant performance improvement from delaying the decision token. This suggests that deciding whether to invoke a tool can be reliably determined from the image–question pair alone, without requiring extended intermediate reasoning. Moreover, the delayed design complicates inference-time control, as enforcing a specific reasoning mode requires multi-stage decoding. Overall, the first-step decision provides a simpler and more practical design without sacrificing performance. We further study the training strategy without explicit <tool_on> / <tool_off> tokens, where the model follows <think> ... </think> <tool_call> ... </tool_call> or <think> ... </think> <answer> ... </answer>. The corresponding results are reported as AutoTool$_{\text{notoken}}$. Similar to AutoTool$_{\text{delay}}$, the decision of whether to invoke a tool is made after the thinking phase. Although the overall performance is comparable to that of our AutoTool, this design remains less flexible at test time as it does not allow direct control over the reasoning mode.

In our method, we do not include a KL regularization term, allowing the model to freely explore and converge faster. For

comparison, AutoTool$_{w/ KL}$ reports results with a KL coefficient of 0.01. Introducing KL restricts the model's exploration, making it harder to learn more optimal reasoning strategies.

## J. Effect of Reward Model Scale

In our main experiments, we adopt Qwen2.5-72B-Instruct (Yang et al., 2024a) as the reward model. To study the impact of reward model capacity, we conduct an ablation using smaller models from the same Qwen2.5-Instruct family, including 32B, 14B, and 7B. The quantitative results are summarized in Table 13. As shown in the results, larger reward models consistently lead to better downstream performance. We attribute this improvement to their stronger ability to provide more accurate feedback for open-ended responses, which is particularly important in reinforcement learning with verifiable rewards. In contrast, smaller reward models tend to produce noisier or less discriminative reward signals, making it harder for the policy to distinguish between subtly different reasoning outcomes. Despite these differences, our method consistently improves performance compared with base model (*i.e.* Qwen2.5-VL-7B (Bai et al., 2025)) across all reward model scales, demonstrating its robustness to the choice of reward model.

*Table 13.* Effect of Reward Model Scale.

| Exp | HRbench-4K | | | HRbench-8K | | | V* | | |
|---|---|---|---|---|---|---|---|---|---|
| | FSP | FCP | Overall | FSP | FCP | Overall | Attribute | Spatial | Overall |
| Base Model | 81.8 | 57.5 | 69.6 | 74.0 | 52.0 | 63.0 | 67.0 | 72.4 | 69.1 |
| AutoTool$_{Reward-7B}$ | 87.5 | 58.8 | 73.1 | 81.0 | 55.0 | 68.0 | 83.5 | 75.0 | 80.1 |
| AutoTool$_{Reward-14B}$ | 91.3 | 59.0 | 75.1 | 86.0 | 58.3 | 72.1 | 88.7 | 81.6 | 85.9 |
| AutoTool$_{Reward-32B}$ | 91.8 | 60.3 | 76.0 | 87.3 | 59.3 | 73.3 | 89.6 | 85.5 | 88.0 |
| AutoTool$_{Reward-72B}$ | **92.5** | **61.3** | **76.9** | **88.0** | **60.0** | **74.0** | **91.3** | **88.2** | **90.1** |

## K. Mode-forced Evaluation

In addition to adaptive tool invocation, our model also allows manually constraining its reasoning behavior by inserting special tokens or prompt instructions that enforce a specific reasoning mode. As shown in Table 14, both the fully tool-assisted (`<tool_on>`) and tool-free (`<tool_off>`) variants achieve competitive performance, demonstrating that each reasoning mode is well trained under our training strategy. The fully tool-assisted mode achieves slightly higher accuracy on certain splits but incurs additional inference overhead. By contrast, the adaptive mode selection of AutoTool achieves the best overall performance by dynamically choosing the most suitable reasoning strategy according to the characteristics of each query. We further explore the effect of forcing AutoTool to use a reasoning mode opposite to its preferred choice at test time. Specifically, we enforce `<tool_on>` for samples where AutoTool originally predicts `<tool_off>`, and vice versa. The corresponding results are reported as AutoTool$_{reverse}$. The majority of samples are forced into unsuitable reasoning modes, leading to the worst overall performance among all variants. This observation further highlights the importance of selecting an appropriate reasoning mode for each instance.

*Table 14.* Mode-forced evaluation results.

| Exp | HRbench-4K | | | HRbench-8K | | | V* | | |
|---|---|---|---|---|---|---|---|---|---|
| | FSP | FCP | Overall | FSP | FCP | Overall | Attribute | Spatial | Overall |
| AutoTool$_{on}$ | **93.3** | 61.3 | **77.3** | 88.0 | 59.8 | 73.9 | 89.6 | **89.5** | 89.5 |
| AutoTool$_{off}$ | 92.3 | 59.0 | 75.6 | 86.5 | 58.3 | 72.4 | 89.6 | 82.9 | 86.9 |
| AutoTool$_{reverse}$ | 91.5 | 58.5 | 75 | 86.3 | 57.5 | 71.9 | 87.8 | 81.6 | 85.3 |
| AutoTool | 92.5 | **61.3** | 76.9 | **88.0** | **60.0** | **74.0** | **91.3** | 88.2 | **90.1** |

## L. Test Performance over Training Progress

To illustrate the evolution of downstream performance during RL training, we report test accuracy on three representative benchmarks (HRBench-4K, HRBench-8K and V*) measured every 10 training steps over the full 80-step schedule. Figure 6 plots the performance curves for each benchmark. Each curve reports the performance of Overall, FSP, and FCP (or Attribute/Spatial for V*) as the model progresses from 10 to 80 training steps. The results demonstrate that AutoTool steadily improves across all benchmarks throughout training.

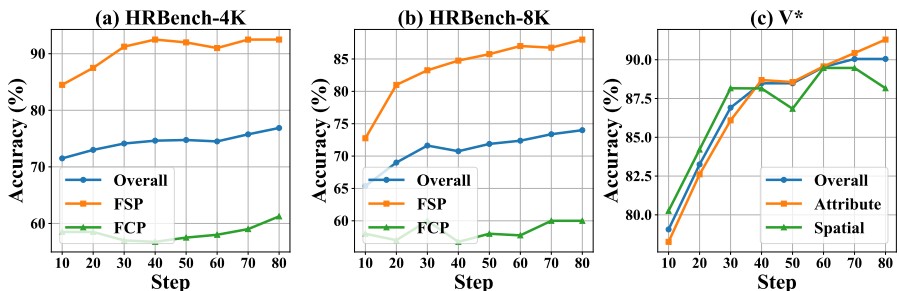

*Figure 6.* Test accuracy across different training steps on HRBench-4K, HRBench-8K, and V*.

## M. Extension to the Multi-tool Setting

Although the main experiments focus on a zoom-in tool for clarity and controlled analysis, our method is not restricted to a single tool type. To evaluate its generality, we further study our method in a multi-tool setting based on Deep-EyesV2 (Hong et al., 2025), a recently proposed framework that supports interleaved invocation of heterogeneous tools, including programmatic code execution and web retrieval, going well beyond simple image cropping.

In DeepEyesV2, multi-tool capabilities are primarily elicited through curated SFT data, which encourages the model to invoke appropriate tools during reasoning. During the subsequent RL stage, only accuracy and format rewards are applied, deliberately avoiding explicit tool-use rewards. While this design partially alleviates excessive tool invocation, the heavy reliance on tool-centric SFT data still biases the model toward frequent tool usage. As a result, the model exhibits a strong preference for invoking tools, and its tool-free reasoning capability remains under-optimized.

*Table 15.* Performance in Multi-tool Setting.

| Exp | Size | Training | HRbench-4K | | | HRbench-8K | | | V* | | |
| | | | FSP | FCP | Inference | FSP | FCP | Inference | Attribute | Spatial | Inference |
|---|---|---|---|---|---|---|---|---|---|---|---|
| DeepEyesV2 | 7B | 50.3 h | 90.5 | 62.0 | 55.75 min | 87 | 60.8 | 63.12 min | 86.1 | 82.9 | 2.62 min |
| DeepEyesV2$_{AMB+MSPO}$ | 7B | **40.4 h** | **92.3** | **62.8** | **37.52 min** | **88.8** | **61.5** | **42.25 min** | **88.7** | **84.2** | **1.82 min** |

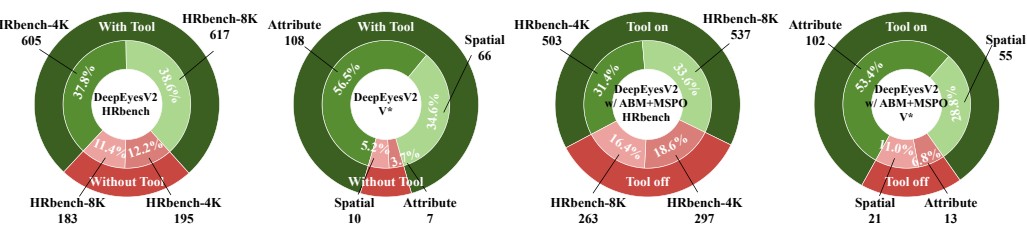

*Figure 7.* The outer ring shows the proportion of the dual reasoning modes on two datasets, while the inner ring presents their distribution across different splits within each dataset. The left two plots correspond to DeepEyesV2, and the right two plots correspond to DeepEyesV2 integrated with AMB and MSPO.

This SFT-then-RL training paradigm shares a closely aligned objective with our AMB: both aim to first establish tool-use competence and subsequently enable more flexible exploration. However, AMB explicitly balances the relative importance of tool-based and tool-free reasoning modes, preventing premature collapse into tool-dominant behaviors. We integrate AMB and MSPO into the DeepEyesV2 training pipeline, demonstrating that our method is plug-and-play with existing reinforcement learning with verifiable rewards (RLVR) algorithms. Quantitative results are reported in Table 15, showing that our approach achieves higher overall task performance while significantly reducing both training and inference overhead.

Figure 7 further illustrates the tool invocation ratios across different benchmarks for DeepEyesV2 with and without AMB+MSPO. The results indicate that our method generalizes well to heterogeneous multi-tool settings and effectively mitigates tool over-reliance beyond the single zoom-in tool studied in the main paper.

## N. More Cases

We provide several representative question–answer examples generated by our AutoTool, covering various task types including perception, hallucination, grounding, and reasoning, as illustrated in Figure 8, Figure 9, and Figure 10. These examples provide qualitative evidence of the model's capability across different dimensions.

We also present several failure cases in Fig. 11. In the first example, which involves counting the number of computers, the model incorrectly assesses the task as simple during mode selection and therefore chooses not to invoke the tool, failing to detect the second computer. In the second example, the model selected the correct reasoning mode but localized the wrong region, resulting in an incorrect answer. The third example is more deceptive. At a glance, the image appears to contain three dogs, making the question seem straightforward. However, careful inspection reveals an additional small white dog located between a black and a yellow dog. In this case, fine-grained visual inspection is required for accurate counting. These failure cases highlight the challenges of reliable reasoning mode selection and precise visual localization.

## O. Limitations and Future Work

Our method explicitly controls whether the model invokes tools in subsequent reasoning by predicting special tokens, and has been effectively validated in both single-tool and multi-tool settings (Section M). However, in more complex scenarios involving sequential tool calls, both redundant and insufficient tool usage may lead to incorrect final answers. Accurately identifying ineffective steps within a tool-call chain therefore becomes a key challenge. A natural direction for future work is to extend the reward formulation to account for tool-chain quality, enabling more fine-grained supervision over the contribution of each tool invocation. Possible strategies include measuring the marginal utility of individual tool calls, or incorporating trajectory-wise evaluation rewards that assess the overall efficiency and coherence of the tool-use sequence. Exploring such reward designs in a stable and scalable manner remains an open problem.

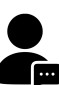

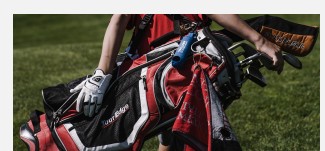

**Question: What piece of equipment is prominently visible in the person's possession?**
**A. A golf bag with golf clubs**
**B. A fishing rod**
**C. A baseball bat**
**D. A tennis racket**

Perception

<tool_off>
<think>Since the question asks about the piece of equipment prominently visible in the person's possession, and the image shows a golf bag with golf clubs, it's clear that a close inspection is not necessary. The equipment is clearly visible and identifiable without the need for a zoom-in tool.</think>
<answer>A. A golf bag with golf clubs</answer>

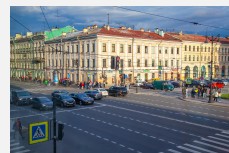

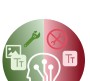

**Question: What is the color of the cyclist's box?**
**A. The color of the cyclist's box is green.**
**B. The color of the cyclist's box is black.**
**C. The color of the cyclist's box is yellow.**
**D. The color of the cyclist's box is red.**

Perception

<tool_on>
<think>Since the question asks for the color of the cyclist's box, it's likely that a close inspection of the image is necessary. Using the image_zoom_in_tool can help us focus on the specific area where the cyclist's box is located, allowing us to verify its color more accurately.</think>
<tool_call>{\"name\": \"image_zoom_in_tool\", \"arguments\": {\"bbox_2d\": [320, 729, 401, 819]}}</tool_call>

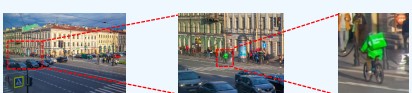

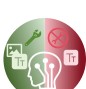

<think>The cyclist's box is green, which matches the color of the box in the zoomed image.</think>
<answer>A. The color of the cyclist's box is green.</answer>

*Figure 8.* Qualitative examples of perception benchmark generated by AutoTool.

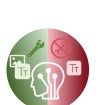

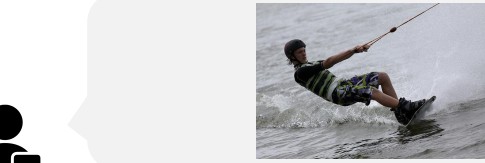

**Question: Is there a car in the image?**

**Hallucination**

<tool_off>
<think>**Upon analyzing the image, there is no car present. The image shows a person water skiing, and there are no vehicles visible.**</think>
<answer>**No**</answer>

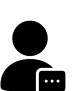

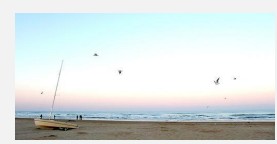

**Question: Is there a person in the image?**

**Hallucination**

<tool_on>
<think>**Since the question asks if there is a person in the image, it's likely that a close inspection of the image is necessary. Using the image_zoom_in_tool can help us focus on the people in the image, which might be small and not clearly visible. This tool will allow us to verify if there are any people present.**</think>
<tool_call>**{\"name\": \"image_zoom_in_tool\", \"arguments\": {\"bbox_2d\": [35, 199, 135, 234]}}**</tool_call>

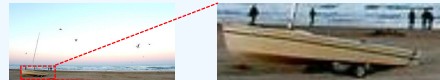

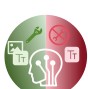

<think>**Upon zooming in, I can see that there are people in the image. The presence of people is confirmed by their distinct shapes and positions on the beach.**</think>
<answer>**Yes, there is a person in the image.**</answer>

*Figure 9.* Qualitative examples of hallucination benchmark generated by AutoTool.

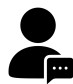

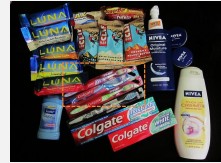

Please find \"\"The blue toothbrush.\", \"The blue handled toothbrush still in its' package.\"\" with bboxs.
Compare the difference between object(s) and find the most closely matched object(s).

Grounding

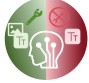

To find the object \"The blue toothbrush.\", \"The blue handled toothbrush still in its' package.\", I need to identify the toothbrushes in the image and determine which one matches the description.

1. **Identify the toothbrushes**: There are two toothbrushes visible in the image. One is blue and the other is red.
2. **Compare the descriptions**: The description specifies a \"blue toothbrush.\" This matches the blue toothbrush in the image.
3. **Locate the blue toothbrush**: The blue toothbrush is the one on the left side of the image, still in its packaging.

Given the above reasoning, the most closely matched object is the blue toothbrush.

<answer>{\"bbox_2d\": [163,188,406,348]}</answer>"

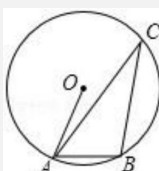

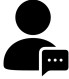

Question: As shown in the figure, A, B, C are three points on ⊙ O, ∠ACB = 25.0, then the degree of ∠BAO is ()
Choices:
(A) 50°
(B) 55°
(C) 60°
(D) 65°

Reasoning

To solve this problem, we need to use the properties of circles and angles.

1. **Identify the given information:**
   - ∠ACB = 25°
   - A, B, and C are points on the circle ⊙ O.
   - O is the center of the circle.

2. **Use the inscribed angle theorem:**
   - The angle subtended by an arc at the center of the circle is twice the angle subtended by the same arc at any point on the remaining part of the circle.
   - Therefore, ∠AOB = 2 * ∠ACB = 2 * 25° = 50°.

3. **Determine ∠BAO:**
   - ∠BAO is an angle in the isosceles triangle OAB (since OA and OB are radii of the circle and thus equal).
   - In triangle OAB, ∠OAB = ∠OBA because OA = OB.
   - The sum of the angles in a triangle is 180°.
   - ∠OAB + ∠OBA + ∠AOB = 180°.
   - Let ∠OAB = ∠OBA = x.
   - Then, 2x + 50° = 180°.
   - Solving for x, we get 2x = 130°, so x = 65°.

Therefore, the degree of ∠BAO is 65°.

The correct answer is <answer>D</answer>.

*Figure 10.* Qualitative examples of grounding and reasoning benchmark generated by AutoTool.

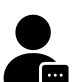
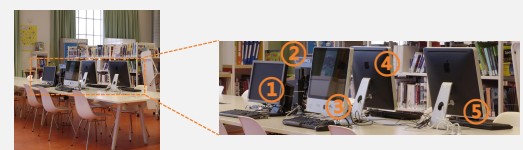
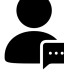
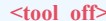

How many computers are visible on the table?
A. Three computers
B. Five computers
C. Two computers
D. Four computers

<tool_off>
<think>Since the question asks for the number of computers visible on the table, and we can directly observe that there are four computers on the table, we don't need to use a tool. We can count the computers directly from the image.</think>
<answer>D. Four computers</answer>  GT: B. Five computers

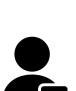
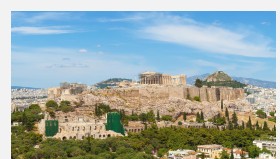

What is the relative position of the pink banner compared to the green trash bin?
A. To the right and slightly in front of the green trash bin
B. Directly above the green trash bin
C. To the left and slightly in front of the green trash bin
D. To the left and slightly behind the green trash bin

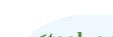

<tool_on>
<think>Since the question asks about the relative position of the pink banner compared to the green trash bin, it's likely that a close inspection of the image is necessary. Using the image_zoom_in_tool can help us focus on the specific areas where the green trash bin and the pink banner are located, allowing us to determine their relative positions more accurately.</think>
<tool_call>{\"name\": \"image_zoom_in_tool\", \"arguments\": {\"bbox_2d\": [500, 1500, 2000, 1950], \"label\": \"green trash bin\"}}</tool_call>

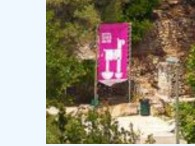
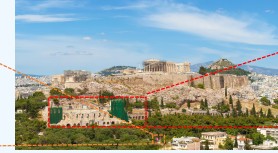
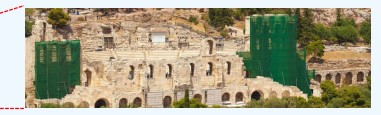

<think>By observing the zoomed image, it appears that the pink banner is to the right and slightly in front of the green trash bin. The green trash bin is covered with a green net, and the pink banner is positioned to the right of it.</think>
<answer>A. To the right and slightly in front of the green trash bin</answer>
 GT: D. To the left and slightly behind the green trash bin

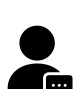
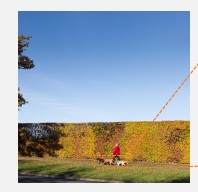
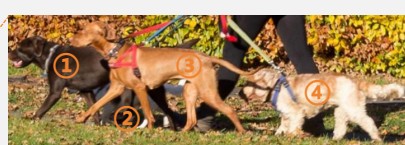

How many dogs are being walked in the image?
A. Four
B. Five
C. Two
D. Three

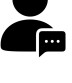

<tool_off>
<think>From the image, it appears that there are three dogs being walked by the person. </think>
<answer>D. Three</answer>  GT: A. Four

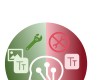

*Figure 11.* Failure cases. Orange boxes denote the ground-truth regions of interest that the model should attend to, while the red boxes show the regions actually selected for zoom-in.

