# OpenReview forum: "Are Tools Always Beneficial? Learning to Invoke Tools Adaptively for Dual-Mode Multimodal LLM Reasoning"
_ICML.cc/2026/Conference — ICML 2026 regular_

### Official Review · Reviewer_Lb1z · 2026-03-08

**Soundness:** 2
**Presentation:** 2
**Significance:** 3
**Originality:** 3
**Overall Recommendation:** 4
**Confidence:** 4

**Summary:**

This paper investigates a significant issue in current tool-use topic of MLLMs, the problems of “overuse” and “blind trust” in tool-calling The authors point out that redundant tool-callings not only increase the computational cost but might also introduce visual noise that leads to incorrect reasoning. To address this issue, the paper proposes the AutoTool framework, which uses reinforcement learning (RL) to train the model to adaptively switch between a direct reasoning mode and a tool-assisted mode. Experiments on the V* benchmark demonstrate that the proposed method improves accuracy while reducing inference overhead.

**Compliance With Llm Reviewing Policy:**

Affirmed.

**Final Justification:**

Authors have resolved my concerns and I decide to increase my final rating to four.

**Key Questions For Authors:**

1. Tool Generalization:
The framework is currently validated only on a visual zoom-in tool. If the system interacts with a search engine that may return incorrect facts, can AutoTool learn an avoidance strategy through RL? Please provide preliminary experiments with other heterogeneous tools.

2. Reward Function Sensitivity:
How sensitive is the model’s accuracy to small changes in the Efficiency Penalty term? Is there a robust range of hyperparameters, or does the method require dataset-specific tuning?

3. Please also response to the above weaknesses.

**Limitations:**

yes

**Strengths And Weaknesses:**

- **Strengths**:

    - Technical Soundness:
      - The experiments compare the trade-off between always calling tools and adaptive calling, and the empirical results support the intuitive conclusion that tools are not always beneficial.
      - To support this conclusion and as a solution, this paper propose two approaches (or "dual-mode" as defined in this paper) to enhance MLLM reasoning regarding efficient tool-use. The balancing issue of tool-calling frequence is addressed by RL with specifically designed rewards.

    - Presentation: The paper clearly motivates the problem, and the figures illustrate concrete failure cases of tool usage.

    - Significance: The paper discusses the efficiency-cost trade off in agentic reasoning, which is an important and practical issue.

    - Originality: The paper introduces the concept of dual-mode adaptive switching.

- **Weaknesses**:
  - Soundness:
    - The experimental setup presents a potential risk of overfitting. The decision logic of AutoTool heavily depends on the design of the RL reward function. In particular, the hyperparameter tuning of the Efficiency Reward has a substantial impact on the results. If this balance is achieved through careful tuning on a specific test set (e.g., V*), the adaptive capability may degrade rapidly when applied to more general open-domain tasks.
    - Besides, the experimental scope is somehow narrow. Most experiments focus almost exclusively on a single tool type, the visual zoom-in. This type of tool is relatively low-risk. The generalizability of the proposed framework to more complex tools, such as external knowledge retrieval (RAG) or code execution, which involve higher hallucination risks and greater complexity, remains unverified.
    - There are some missing references, indicating that the authors did not conduct thorough research of the backgroung and related works of this topic, including but not limited to:
      - [1] focuses on mathematical reasoning models and explores how models dynamically decide whether to integrate code tools through metacognitive decision-making. This type of metacognitive decision mechanism is conceptually very similar to the dual-mode switching proposed in AutoTool. The paper should clarify what fundamental algorithmic innovation its RL reward design provides compared with the more established Expectation-Maximization (EM) framework in a multimodal context.
      - The “adaptive tool calling” discussed in this paper is essentially a form of perception-driven decision-making by the model. ActiView [2] specifically focus on whether MLLMs can actively perform actions (e.g., Zoom-in and -out) based on their current understanding in order to obtain key information. If ActiView has already demonstrated that current models exhibit a substantial gap in active perception abilities, then the modest improvement reported by AutoTool on the V* benchmark may simply reflect overfitting to relatively simple tasks, rather than addressing the fundamental challenges of active perception.
      - [3] proposes an interleaved planning mechanism that allows models to dynamically decide when to invoke tools during reasoning, while supporting parallel execution. The dual-mode switching in AutoTool appears somewhat coarse-grained (either respond directly or rely entirely on tools). Compared with the fine-grained dynamic interleaving used in SPRINT [3], the AutoTool design may appear less flexible when handling complex tasks.

  - Presentation:
    - This paper may not be carefully written, for example, there are mixed usage of quotes, where in line 106-109, authors use single quotes ('xxx') for the main quotation, while they use double quotes ("xx") in the rest, including cases and prompts.
    - Key technical details are described somewhat vaguely, except the general RL formulations. The internal triggering mechanism of the dual-mode switching lacks thorough ablation analysis. For example, it is unclear whether the model’s decision is based on image ambiguity, instruction complexity, or certain hidden feature representations. The paper lacks interpretability analysis to clarify this decision process.


Missing references:

[1] Wang, H., Li, L., Qu, C., Xu, W., Zhu, F., Chu, W., & Lin, F. _To code or not to code? adaptive tool integration for math language models via expectation-maximization_. ACL 2025 Findings.

[2] Wang, Z., Chen, C., Luo, F., Dong, Y., Zhang, Y., Xu, Y., ... & Liu, Y. (2025, July). _ActiView: Evaluating active perception ability for multimodal large language models_. ACL 2025.

[3] Biju, E., Talaei, S., Huang, Z., Pourreza, M., Mirhoseini, A., & Saberi, A. _Sprint: Enabling interleaved planning and parallelized execution in reasoning models._ NeurIPS 2025.

---

> ### Author Rebuttal · Authors · 2026-03-30
>
> Thank you for your thorough review and for underscoring several positive aspects of this work. Please find below responses to your comments. The corresponding revisions will be incorporated into the revised manuscript.
>
> > Q1/W2. Generalization to Multi-Tool Settings
>
> We evaluate multi-tool generalization based on DeepEyesV2. Please refer to  **W1. Generalization to Multi-Tool Settings in Reviewer 9wCH** for details.
>
> Importantly, retrieval introduces noisy or incorrect outputs. Under our RL formulation, such unreliable tool usage is naturally penalized through lower final rewards, encouraging the model to avoid unhelpful tool calls and rely on internal reasoning when appropriate. These results show that AutoTool generalizes to heterogeneous tools, including scenarios with imperfect external signals, rather than being limited to low-risk visual tools.
>
> > Q2/W1. On reward sensitivity and potential overfitting
>
> We only perform limited tuning on the mode-balancing removal step and inherit all other hyperparameters from DeepEyes. The consistent performance improvements across multiple benchmarks spanning different categories suggest that the observed gains are not due to dataset-specific overfitting, but rather arise from the general effectiveness of the proposed training framework.
>
> The efficiency penalty $R_{tool} = -0.5$ is introduced to discourage unnecessary tool usage, especially when it leads to incorrect answers. To evaluate its sensitivity, we conduct additional experiments by varying this penalty in \{0, -0.2, -0.5, -0.8\}.
>
> |penalty|HRbench-4K|HRbench-8K|V\*|
> |-|-|-|-|
> |0|76.8|73.2|89.5|
> |-0.2|76.6|73.5|89.5|
> |-0.5 (default)|76.9|74.0|90.1|
> |-0.8|77.0|73.6|90.1|
>
> Empirically, performance varies only marginally across these settings, indicating a robust operating range.
>
> > W3. Missing References and Methodological Comparisons
>
> AutoTool is not based on designing a tool-necessity classifier via direct supervision, but on expanding the solution space through dual-mode RL exploration, enabling the model to self-direct appropriate reasoning strategies.
>
> (1) Comparison with the EM.
> While both share the idea of decision-making, EM relies on an offline E-step with trajectory sampling to estimate a latent policy. AutoTool performs on-policy RL with a unified objective, dynamically learning the tool-selection policy without separate sampling. Moreover, AutoTool targets multimodal scenarios with explicit penalties for redundant processing, which is not addressed in EM.
>
> (2) Generalization and Adaptive Perception.
> ActiView highlights the gap in active perception, while AutoTool proposes an RL-based framework to improve it. We note that our training data from V\* is comprised of general vision datasets (*e.g.*, GQA, LVIS, and Objects364), which differ from the V\* test distribution (high-resolution SA-1B images), reducing the risk of overfitting. In addition, AutoTool shows consistent improvements on other benchmarks, and exhibits adaptive switching behavior rather than collapsing to a single mode.
>
> (3) Comparison with SPRINT.
> While both improve efficiency, they differ in objectives and settings. First, SPRINT focuses on fine-grained interleaving in text-only math reasoning for latency reduction via parallelization, whereas AutoTool targets multimodal reasoning and learns when to invoke tools. Second, SPRINT relies on supervised interleaved data, while AutoTool is trained without step-level annotations. In addition, fine-grained interleaving in multimodal scenarios incurs extra visual processing at each step, increasing computational cost and causing potential error accumulation from intermediate feedback. AutoTool offers a more robust and efficient alternative.
>
> We will incorporate these discussions into the revised manuscript.
>
> > W4. Quotation Style Consistency
>
> We will revise the manuscript to ensure a consistent quotation style throughout for improved presentation.
>
> > W5. Interpretability of Mode Selection
>
> We will include more technical details in the revision. While the decision boundary is implicitly learned, we provide empirical evidence to characterize it. Controlled experiments (Table 12 in Appendix K) show clear performance differences under forced modes, indicating meaningful task-dependent decisions. We also observe consistent patterns (Figure 3): tool use is more frequent in high-resolution perception tasks (*e.g.*, V\*), while text-only reasoning is preferred for simpler or globally answerable questions (*e.g.*, POPE), suggesting adaptation to visual complexity  and task difficulty.
>
> Qualitative evidence from `<think>...</think>` traces further reveals interpretable cues: the model avoids tools when close inspection is unnecessary (*e.g.*, “the image shows … and close inspection is unnecessary”) and invokes them when key details are not visible (*e.g.*, “... is not visible in the general view … requiring a close-up”), indicating sensitivity to image ambiguity and instruction complexity.

---

> > ### Author Rebuttal · Reviewer_Lb1z · 2026-04-02
> >
> > Thanks for the reply. I will increase my rating, and hope to see the discussion be included in the final revision.

---

> > > ### Author Response · Authors · 2026-04-03
> > >
> > > We are pleased to hear that our rebuttal has addressed your concerns, and we sincerely appreciate your decision to raise the score. We are also very grateful for your constructive comments and questions, which have played an important role in improving our manuscript. We will incorporate the relevant discussions into the final version.

---

### Official Review · Reviewer_9wCH · 2026-03-11

**Soundness:** 4
**Presentation:** 3
**Significance:** 3
**Originality:** 2
**Overall Recommendation:** 5
**Confidence:** 4

**Summary:**

This paper investigates the inefficiency and unreliability caused by indiscriminate tool usage in multimodal large language models (MLLMs), where existing methods often invoke tools redundantly or inappropriately, increasing computational overhead and hallucinations. The authors investigate a general area of adaptive multimodal reasoning by proposing AutoTool , a reinforcement learning (RL) framework that dynamically balances tool-assisted and text-centric reasoning modes based on query characteristics. AutoTool introduces two key innovations:
- Mode-Specific Policy Optimization (MSPO) , which applies distinct reward functions to train the model for accurate tool utilization in complex tasks and pure textual reasoning in simpler cases
- Adaptive Mode Balancing (AMB) , which adjusts reward coefficients during training to ensure sufficient exploration of both modes before relaxing constraints for autonomous mode selection.
Experiments on benchmarks like V, HRbench, and POPE demonstrate that AutoTool achieves higher accuracy (e.g., 21.8% gain on V ) and efficiency (44.9% faster than DeepEyes) while reducing unnecessary tool operations. The method generalizes to multi-tool settings and avoids the pitfalls of prior approaches that rely on fixed tool orchestration or proxy signals, offering a flexible and interpretable solution for MLLM reasoning.

**Compliance With Llm Reviewing Policy:**

Affirmed.

**Final Justification:**

Final Recommendation: 5: accept

The rebuttal thoroughly addresses all key concerns with additional experiments, mechanistic explanations, and empirical evidence. AutoTool’s adaptive reasoning framework now demonstrates robustness across diverse tool types and hyperparameter ranges, making it a valuable contribution to MLLM tool orchestration. The paper’s practical impact, combined with its refined theoretical and empirical grounding, warrants strong endorsement.

**Key Questions For Authors:**

- The paper lacks formal theoretical analysis of why the dual-mode RL framework ensures optimal tool usage. Could the authors provide convergence guarantees or theoretical insights into how the GRPO algorithm with mode-specific rewards avoids suboptimal policies (e.g., tool overuse or underuse)?
- AMB tends to keep the <tooloff> and <toolon> modes balanced, which also requires the training data itself to be relatively balanced; otherwise, it could harm the training effectiveness. It is unclear what the authors did regarding the balance of the training data.
- The AMB strategy dynamically adjusts λmode to balance exploration and exploitation. Is there empirical evidence (e.g., training curves) showing that this avoids local optima or premature bias toward one mode? How sensitive is convergence to λmode hyperparameters?
- Regarding Mode-specific tool reward, when will Rtool=0 happen in <tool_on> mode?

**Limitations:**

yes

**Strengths And Weaknesses:**

Strengths:
- The paper demonstrates technical soundness through its empirical approach. The proposed AutoTool framework is rigorously evaluated on multiple benchmarks (V*, HRbench, POPE, Reasoning datasets), with ablation studies (Tables 4–6) validating the necessity of key components like MSPO and AMB.
- The reinforcement learning (RL) framework is well-structured, leveraging GRPO with mode-specific rewards to train the model to adaptively choose between tool-assisted and text-centric reasoning. The reward design (accuracy, format compliance, and mode-specific tool usage) aligns with the goal of balancing efficiency and reliability.
- AutoTool achieves state-of-the-art results on multiple benchmarks, demonstrating practical utility.

Weaknesses:
- The experiments focus on a single tool (zoom-in), and the multi-tool extension (Appendix M) is briefly discussed without extensive validation. The method’s generalization to diverse tool types (e.g., OCR, segmentation) remains unproven at scale.
- While the supplementary material mentions prompt designs, the exact system/user prompts (Appendix A) and reward model thresholds are not fully specified, potentially hindering replication.

---

> ### Author Rebuttal · Authors · 2026-03-30
>
> We are grateful for the reviewer’s constructive remarks and suggestions. We respond to each point in the following, and the corresponding revisions will be incorporated into the revised paper.
>
> > Q1. Theoretical justification and convergence behavior
>
> Our framework incorporates several mechanisms that theoretically mitigate suboptimal policies, as detailed below.
>
> (1) Stable optimization via GRPO:
> We optimize a PPO-style clipped objective, ensuring bounded policy updates and inheriting the empirical stability of policy gradient methods.
>
> (2) Structured dual-mode policy:
> Modeling $\pi_\theta(a|x)=\sum_m \pi_\theta(m|x)\pi_\theta(a|x,m)$ decomposes decision-making into mode selection $m$ and action selection. This two-level policy reduces optimization difficulty and avoids collapse to a single mode.
>
> (3) Mode-specific rewards prevent degeneracy:
> Rewards $R_{\text{tool}}$ provide mode-conditional signals that penalize ineffective tool usage while rewarding correct reasoning, directly discouraging degenerate policies.
>
> (4) Adaptive exploration via AMB:
> We dynamically adjust $\lambda_{\text{tool}}^{\text{mode}}$ to ensure sufficient exploration, preventing premature convergence to a single mode.
>
> Empirically, we observe stable training, balanced mode exploration, and adaptive mode selection at inference, indicating convergence to a non-degenerate policy. A formal convergence analysis for dual-mode RL remains an interesting direction for future work.
>
> > Q2. Effect of data balance on AMB
>
> We do not perform any explicit data balancing and directly use the original DeepEyes training set. Importantly, AMB does not rely on balanced data; instead, it mitigates policy bias (*e.g.*, preference for `<tool_off>`) rather than enforcing data-level balance. Specifically, AMB dynamically adjusts reward coefficients based on mode frequency during training, encouraging exploration of underrepresented modes regardless of the data distribution. This policy-level mechanism remains effective under imbalanced data. Empirically, we observe stable training and adaptive mode selection across benchmarks, suggesting that AMB does not require explicitly balanced data.
>
> > Q3. Empirical evidence and sensitivity of AMB
>
> We provide empirical evidence in Fig. 1(c), which shows the evolution of tool usage during training. The model maintains balanced exploration between modes without collapsing, and the distribution gradually stabilizes, indicating that AMB avoids premature bias and local optima.
>
> Regarding $\lambda^{mode}$, the coefficient $\lambda_{tool}^{mode}$ is not fixed but dynamically adjusted based on mode frequency, automatically compensating for imbalance and reducing mode collapse. For $\lambda^{base}_{tool}$, we use the default value 1.2 and evaluate sensitivity over \{1.0, 1.2, 1.4\}.
>
> |$\lambda^{base}_{tool}$|HRbench-4K|HRbench-8K|V\*|
> |-|-|-|-|
> |1.0|76.0|73.9|89.5|
> |1.2 (default)|76.9|74.0|90.1|
> |1.4|77.3|73.5|90.1|
>
> Performance remains stable with minimal variation, suggesting that AMB is not sensitive to this hyperparameter and does not require careful tuning.
>
> > Q4. Clarification on $R_{\text{tool}}$ in the `<tool_on>` mode
>
> In the `<tool_on>` mode, $R_{tool} = 0$ when tool execution fails. This includes cases where (1) no valid tool call is generated, or (2) the invocation is malformed and cannot be parsed or executed. This design enforces correct tool usage beyond format reward $R_{format}$, which only ensures the overall response structure but does not guarantee valid and executable tool calls.
>
> > W1. Generalization to Multi-Tool Settings
>
> We evaluate multi-tool generalization based on DeepEyesV2, which supports heterogeneous tools (perception, computation, retrieval). By integrating AMB and MSPO, we observe consistent performance gains, along with significantly reduced training and inference overhead.
>
> |Exp|Size|Training|HRbench-4K (Overall)|HRbench-4K (Inference)|HRbench-8K (Overall)|HRbench-8K (Inference)|V\* (Overall)|V\* (Inference)|
> |-|-|-|-|-|-|-|-|-|
> |DeepEyesV2|7B|50.3 h|76.3|55.75 min|73.9|63.12 min|84.8|2.62 min|
> |DeepEyesV2$_{AMB+MSPO}$|7B|40.4 h|77.5|37.52 min|75.1|42.25 min|86.9|1.82 min|
>
> While OCR or segmentation tools are not explicitly included, they typically require dedicated SFT data for tool-specific capabilities, which is orthogonal to our focus. AutoTool is designed to learn *when* to use tools rather than *how* to use them, and can thus be naturally extended to additional tool types.
>
> > W2. Prompt and Reward Specification
>
> The full system and user prompts are provided in Appendix A. For the reward model, we do not use continuous scores or thresholds. The rewards are directly given as discrete judgments (0/1) from the judge model (*e.g.*, Qwen2.5-72B-Instruct).

---

> > ### Author Rebuttal · Reviewer_9wCH · 2026-04-01
> >
> > For Q3, the authors provide a table and figure showing minimal performance variation and balanced mode exploration. However, the sensitivity analysis is limited to a narrow hyperparameter range and lacks deeper discussion (e.g., failure modes for extreme λ values).
> >
> > For W1, the rebuttal cites experiments on DeepEyesV2 with heterogeneous tools and performance gains. However, the absence of OCR/segmentation tools (which are common in real-world applications) is not fully addressed.

---

> > > ### Author Response · Authors · 2026-04-03
> > >
> > > We sincerely thank the reviewer for the comments, which provide valuable guidance for further refining our work. We concur with these suggestions and have provided detailed responses to each point below. We will incorporate the corresponding updates into the revised manuscript.
> > >
> > > > For Q3, the authors provide a table and figure showing minimal performance variation and balanced mode exploration. However, the sensitivity analysis is limited to a narrow hyperparameter range and lacks deeper discussion (e.g., failure modes for extreme λ values).
> > >
> > > We thank the reviewer for this suggestion.  In addition to the default range, we further evaluate extreme values of $\lambda^{base}_{tool}$ to analyze failure modes.
> > >
> > > |$\lambda^{base}_{tool}$|HRbench-4K|HRbench-8K|V\*|
> > > |-|-|-|-|
> > > |0.0|72.4|69.8|84.3|
> > > |0.5|75.5|72.8|88.5|
> > > |1.0|76.0|73.9|89.5|
> > > |1.2 (default)|76.9|74.0|90.1|
> > > |1.4|77.3|73.5|90.1|
> > > |3.0|75.0|72.1|88.0|
> > > |5.0|71.4|68.4|83.8|
> > >
> > > As shown in the table, the method achieves stable performance around the default value (1.2), and remains robust within a moderate range (0.5–3.0). Beyond this range, we observe clear degradation due to reward imbalance.
> > >
> > > When $\lambda_{tool}^{base}$ is too small, the contribution of $R_{tool}$ becomes negligible, weakening supervision on tool usage behavior and leading to reward hacking. For example, when $\lambda_{tool}^{base}=0.0$, the model tends to select the `<tool_on>` mode ($\sim$80\%), **but performs pure text reasoning without valid tool calls**, exploiting $R_{tool}^{on}=0$ while benefiting from a larger $\lambda_{tool}^{off}$.
> > >
> > > When $\lambda_{tool}^{base}$ is too large, the relative difference between $\lambda_{tool}^{on}$ and $\lambda_{tool}^{off}$ diminishes, biasing the policy toward the base model preference (`<tool_off>`). For instance, when $\lambda_{tool}^{base}=5.0$, **the model collapses to pure text reasoning**. In addition, **the imbalance between reward terms negatively affects adherence to the expected output format** (`<think>...</think><answer>...</answer>`).
> > >
> > > Overall, $\lambda_{tool}^{base}\approx1.2$ provides a good balance between performance and behavior. Within a moderate range (0.5–3.0), the model maintains adaptive tool usage, while extreme values lead to mode collapse and degraded reasoning quality.
> > >
> > > >  For W1, the rebuttal cites experiments on DeepEyesV2 with heterogeneous tools and performance gains. However, the absence of OCR/segmentation tools (which are common in real-world applications) is not fully addressed.
> > >
> > > We agree that OCR and segmentation are important in real-world applications.  To evaluate generalization to such tools, we extend our method on top of V-Tool RL[1], which supports a diverse set of chart-oriented tools, including spatial localization (POINT), visual annotation (e.g., drawing reference lines), region-level **segmentation** and zooming, as well as **OCR** for extracting textual elements. Following the official implementation, we conduct training with our modules on chart understanding tasks and evaluate on a held-out subset of 1,000 samples from CHARTGEMMA[2]. The results are shown below:
> > >
> > > |Method|Accuracy|Avg. Tool Num|
> > > |-|-|-|
> > > |V-Tool RL|59.4|0.097|
> > > |V-Tool RL$_{AMB+MSPO}$|60.9|0.377|
> > >
> > > We observe that the V-Tool RL trained with only accuracy and format rewards, exhibits clear mode collapse (tool usage $\sim$0.1). In contrast, our method expands the solution space through dual-mode RL exploration, leading to both improved accuracy and more reasonable tool usage frequency. These results suggest that AutoTool generalizes effectively to OCR- and segmentation-based tool settings.
> > >
> > > [1] Su, Zhaochen, et al. Openthinkimg: Learning to think with images via visual tool reinforcement learning. arXiv  2025.
> > >
> > > [2] Masry, Ahmed, et al. Chartgemma: Visual instruction-tuning for chart reasoning in the wild. Proceedings of the 31st International Conference on Computational Linguistics: Industry Track. 2025.

---

### Official Review · Reviewer_Zn1e · 2026-03-13

**Soundness:** 2
**Presentation:** 3
**Significance:** 2
**Originality:** 2
**Overall Recommendation:** 4
**Confidence:** 3

**Summary:**

This paper investigates adaptive tool invocation in multimodal LLM reasoning. The authors argue that always invoking tools can introduce unnecessary computation and potentially degrade prediction quality. They propose AutoTool, which introduces two explicit reasoning modes (<tool_on> and <tool_off>) and trains them using reinforcement learning with mode-specific rewards and an adaptive mode-balancing mechanism. Experiments on perception, grounding, hallucination, and reasoning benchmarks demonstrate improvements over the base Qwen2.5-VL-7B model and over DeepEyes on several tasks, while also reducing training and inference time. The central claim is that the method enables models to learn when tool usage is beneficial, in addition to learning how to use tools.

**Compliance With Llm Reviewing Policy:**

Affirmed.

**Final Justification:**

In the authors' rebuttal, they provide detailed experimental results to address my concerns. Thus, I decide to raise my scores.

**Key Questions For Authors:**

1. Why is the routing decision implemented using special tokens (<tool_on>, <tool_off>) instead of natural language instructions?
Since modern LLMs have strong instruction-following ability, it would be interesting to know whether natural-language routing would work equally well.

2. Would the same method apply to text-only tool use?

**Limitations:**

Yes

**Strengths And Weaknesses:**

**Strengths:**
1. The empirical evaluation is relatively comprehensive. The paper evaluates the method on multiple benchmarks covering perception, grounding, hallucination, and reasoning tasks. Experimental details are also reasonably well documented, including training setup, rollout configuration, and reward design.
2. The paper studies tool invocation for multimodal reasoning, which is an important problem. In addition, balancing tool usage to avoid unnecessary tool calls while maintaining reasoning accuracy is an important issue for efficiency and scalability in tool-augmented systems.
3. The proposed method is conceptually simple and easy to understand. The work also addresses an emerging topic—adaptive tool invocation in multimodal reasoning—which is becoming increasingly relevant as multimodal LLMs adopt external tools for perception and reasoning.


**Weaknesses:**
1. Lack of ablations on prompt and token dependence.
The proposed method relies heavily on special control tokens (<tool_on> and <tool_off>) to represent the tool-selection decision. However, the paper does not provide ablations that test the robustness of the approach when these tokens or the associated prompting scheme are modified or removed. For example, it would be useful to evaluate whether the model still performs well when the system prompt does not explicitly enforce this token-based routing.

2. It is unclear whether the improvements come from better tool selection or from reward shaping.
The paper claims that the main contribution is learning when to use tools. However, the experiments do not isolate the effect of the decision mechanism itself. A useful control experiment would be to compare against a pipeline where a separate model (e.g., Qwen2.5-VL) predicts whether tool usage is required, and then routes the query either to a tool-based solver (e.g., DeepEyes) or to a standard multimodal model. Such an experiment would help clarify whether the gains come from the learned decision policy or from the overall training strategy.

3. The necessity of the special-token design is not well justified.
The tool-selection decision is implemented via explicit special tokens. It would be helpful to understand whether the same behavior could be learned using natural language instructions instead of dedicated control tokens. Exploring such alternatives could provide insight into whether the improvement is due to the architectural design or simply the prompting format.

4. While the problem of adaptive tool invocation in multimodal reasoning is interesting, it would be helpful to clarify why the proposed approach is specifically tailored to the multimodal "Thinking with Images" setting, rather than being a general solution to tool usage in LLM systems.In fact, controlling the number of tool calls is already a well-studied problem in text-based tool use and LLM agents. Many existing works address similar challenges such as avoiding redundant tool calls, improving tool-selection policies, or optimizing tool usage efficiency. The paper does not clearly articulate what distinguishes the multimodal case from the text-only setting in a fundamental way. Consequently, it remains unclear whether the proposed approach provides new insights specific to multimodal reasoning, or whether it could be directly applied to text-only tool-use scenarios. A discussion or experiment demonstrating how the method differs from, or generalizes beyond, existing text-based tool-selection approaches would strengthen the significance of the work.

5. While the paper studies tool invocation in multimodal reasoning, it is not entirely clear how the proposed approach fundamentally differs from existing work on tool selection and tool usage control in text-based LLM agents. The paper would benefit from a clearer discussion of what new insights arise specifically from the multimodal setting, and how the proposed framework differs from or advances prior work in text-based tool-use optimization.

---

> ### Author Rebuttal · Authors · 2026-03-30
>
> Thank you for your insightful review and valuable suggestions. We address the concerns below and will incorporate the corresponding revisions into the paper.
>
> > Q1/W3. On the Necessity of Special Tokens.
>
> We experiment with replacing explicit control tokens with natural language instructions, allowing the model to implicitly decide tool usage from the input.
>
> Under the same GRPO training setup with only accuracy and format rewards, the model quickly collapses to a text-only reasoning policy. When we further introduce auxiliary rewards to encourage tool usage, the model instead collapses to an always-use-tool policy, indicating unstable training dynamics. These observations suggest that natural language instructions alone are insufficient to reliably learn an adaptive tool-use policy. In contrast, explicit special tokens provide a clear and structured interface for decision-making, enabling more stable credit assignment and effective optimization. Please further refer to **W1. Ablation on Prompt and Token Dependence.**, where explicit tokens consistently yield more reliable performance.
>
> > Q2/W4/W5. Multimodal vs. Text-only Tool Use.
>
> We agree that AutoTool is general and can be applied to text-only tool-use scenarios. However, its impact is more pronounced in multimodal settings due to key differences. First, multimodal tool usage incurs substantially higher computational cost, as operations such as zooming or cropping significantly increase visual tokens, whereas text-based tools (*e.g.*, calculators or code execution) introduce minimal overhead. This makes adaptive tool calls more critical in MLLMs. Second, multimodal reasoning involves a different type of uncertainty. Text-based tools primarily address knowledge gaps, whereas multimodal tools address perception gaps. In MLLMs, excessive or unnecessary visual information can introduce noise and distract the reasoning process. This makes the trade-off between tool usage and direct reasoning more delicate and harder to optimize.
>
> These characteristics motivate our design: instead of relying on fixed decision rules, AutoTool learns a self-directed policy that balances both reasoning modes and selects appropriate strategies under different conditions. Compared to text-based approaches, AutoTool differs in its training paradigm. Methods such as MeCo[1] rely on inference-time signals to trigger tool usage without optimizing the underlying policy, while AutoCode[2] depends on offline data construction and static supervision. In contrast, AutoTool directly optimizes a unified policy via reinforcement learning, enabling joint improvement of tool selection and reasoning through dynamic exploration. Overall, while our framework is general, it is specifically designed to address the high-cost, high-redundancy, and perception-driven challenges unique to multimodal reasoning.
>
> > W1. Ablation on Prompt and Token Dependence
>
> We conduct additional ablations to evaluate the dependence of our method on special control tokens. We remove explicit mentions of these tokens from the system prompt, allowing the model to generate either `<think>...</think><tool_call>...</tool_call>` or `<think>...</think><answer>...</answer>`. Tool usage is then inferred from the output format and used for reward routing, which is functionally equivalent to token-based routing.
>
> |Method|HRbench-4K|HRbench-8K|V\*|
> |-|-|-|-|
> |AutoTool$_{notoken}$|76.5|73.5|89.5|
> |AutoTool|76.9|74.0|90.1|
>
> The results (AutoTool$_{notoken}$) show that the model remains functional without explicit tokens, with only a slight performance drop. This suggests that the approach does not critically depend on explicit token prompting. However, explicit control tokens provide clearer reward routing, avoid heuristic parsing, and enable finer-grained control over tool usage at inference time, leading to a more practical and reliable framework.
>
> > W2. Routing-then-Reasoning
>
> To isolate the effect of the decision mechanism, we implement a Routing-then-Reasoning baseline. Specifically, Qwen2.5-VL-7B first predicts whether zoom-in tools are required, and the query is then routed to either a tool-based solver (DeepEyes) or a standard multimodal model (Qwen2.5-VL-7B).
>
> |Method|HRbench-4K|HRbench-8K|V\*|POPE|
> |-|-|-|-|-|
> |Route|70.5|63.5|69.6|87.2|
> |AutoTool|76.9|74.0|90.1|88.9|
>
> This pipeline underperforms AutoTool across all benchmarks by a clear margin. This indicates that separating decision and reasoning into independent components is insufficient to achieve strong performance. Moreover, the additional routing stage introduces extra overhead, whereas AutoTool integrates decision-making and reasoning within a single model, resulting in a more efficient and practical solution.
>
> [1] Li, Wenjun, et al. Adaptive tool use in large language models with meta-cognition trigger. ACL 2025.
>
> [2] Wang, Haozhe, et al. To code or not to code? adaptive tool integration for math language models via expectation-maximization. ACL 2025 Findings.

---

### Official Review · Reviewer_9mFq · 2026-03-14

**Soundness:** 3
**Presentation:** 3
**Significance:** 3
**Originality:** 3
**Overall Recommendation:** 4
**Confidence:** 4

**Summary:**

This paper challenges the prevailing assumption in tool-augmented multimodal reasoning that tool invocation is always beneficial. The authors observe that indiscriminate tool usage (e.g., zoom-in operations) in multimodal large language models (MLLMs) incurs substantial training/inference overhead and may introduce distracting or misleading visual information, potentially exacerbating hallucinations. To address this, the paper proposes **AutoTool**, a reinforcement learning framework that enables the model to adaptively decide whether to invoke external tools on a per-query basis.

AutoTool introduces three key components: (1) **Explicit dual-mode reasoning** via special control tokens (`<tool_on>` / `<tool_off>`) generated at the first decoding step, routing subsequent reasoning into tool-augmented or text-only trajectories; (2) **Mode-Specific Policy Optimization (MSPO)**, which applies asymmetric reward functions to different reasoning modes — notably penalizing tool invocation when it leads to incorrect answers ($R_{\text{tool}} = -0.5$); and (3) **Adaptive Mode Balancing (AMB)**, which dynamically adjusts reward coefficients based on the empirical mode frequency within each training batch to prevent premature collapse toward the easier `<tool_off>` mode, with the constraint relaxed in later training stages.

The model is built on Qwen2.5-VL-7B and trained with GRPO for 80 iterations on 8 H200 GPUs, using Qwen2.5-72B-Instruct as an online reward model. Experiments span perception (V\*, HRBench), grounding (RefCOCO series, ReasonSeg), hallucination (POPE), and reasoning (MathVista, MathVerse, etc.) benchmarks. AutoTool reports consistent improvements over the base model and DeepEyes, with notable gains on V\* (+21 points overall vs. base) and significant training/inference speedups (20.3% training time reduction, up to 44.9% inference speedup on POPE).

**Compliance With Llm Reviewing Policy:**

Affirmed.

**Key Questions For Authors:**

1. **Reward model sensitivity and reproducibility:** Can you provide (a) the exact prompt template and sampling parameters used for the online reward model judge, (b) the agreement rate between the judge and human evaluation on a held-out subset, and (c) results using a non-Qwen reward model (e.g., from a different model family) to test whether the gains are robust to judge choice? This would significantly improve confidence in reproducibility.

2. **Mode decision quality evaluation:** Beyond end-task accuracy, is there a way to directly evaluate the quality of the tool/no-tool decision? For instance, one could construct a "necessity oracle" using human annotations or proxy signals (e.g., whether the answer changes when the tool is available vs. not) and measure the alignment between AutoTool's decisions and this oracle. This would provide more direct evidence for the claimed adaptive capability.

3. **Multi-tool generalization depth:** In the DeepEyesV2 extension (Appendix M), how does the dual-mode decision interact with different tool types? Does the model learn different invocation patterns for perception tools vs. computational tools? More detailed analysis (e.g., per-tool invocation rates, performance breakdown by tool type) would strengthen the generalizability argument.

4. **Handling sequential tool chains:** The current MSPO rewards only the final answer. For multi-step tool chains, have you considered trajectory-level efficiency rewards (e.g., penalizing trajectory length or incorporating marginal utility of each tool call)? Even preliminary results or theoretical analysis would address the acknowledged limitation in Appendix O.

**Limitations:**

Partially. The authors discuss the technical limitation of sequential tool chain optimization in Appendix O and provide honest failure cases in Appendix N, which is commendable.

**Strengths And Weaknesses:**

### Strengths

**S1. Well-motivated and practically important problem.** The observation that tool invocation is not always beneficial is grounded in clear empirical evidence. Figure 1 effectively illustrates cases where forced zoom-in disrupts global scene understanding (e.g., spatial relationship questions), leading to incorrect answers. The efficiency dimension — that redundant tool calls waste 44.9 training hours and significantly inflate inference time — is directly relevant to real-world deployment.

**S2. Clean and principled method design.** The dual-mode formulation elegantly reduces the adaptive tool-use problem to a discrete binary decision at generation time, which is trainable end-to-end via RL. The MSPO reward structure is well-reasoned: the $R_{\text{tool}} = -0.5$ penalty for incorrect tool-augmented answers directly encodes the cost-awareness missing from prior work like DeepEyes, which rewards tool invocation regardless of answer correctness. The AMB mechanism addresses a real RL training challenge (mode collapse due to the foundation model's text-centric bias), and the ablation in Table 5 convincingly shows that both the exploration phase and the later free-exploration phase are necessary (best results at step-60 removal, 90.1% on V\*).

**S3. Comprehensive and strong experimental results.** The evaluation covers four task categories (perception, grounding, hallucination, reasoning) across 13+ benchmarks, with comparisons against both proprietary (GPT-4o, o3) and open-source models of varying sizes. The wall-clock efficiency comparison (Table 6) is particularly valuable, as it grounds the "efficiency" claim in actual compute savings rather than proxy metrics. The ablation studies (Tables 4, 5, 10, 11, 12) systematically validate each component: MSPO penalty (+1.5 V\* overall), AMB (+5.3 V\* overall), reward model scale sensitivity, mode-forced evaluation, and alternative design choices (delayed decision, no explicit token, SFT warmup, KL regularization). AutoTool-7B outperforms models up to 5x larger (e.g., V\* Overall: 90.1 vs. Qwen2.5-VL-32B's 79.1), demonstrating that adaptive perception can be more effective than brute-force parameter scaling for high-resolution visual understanding. The grounding results (Table 2) show consistent improvements across all RefCOCO splits, suggesting the MSPO penalty also back-pressures the model toward more precise bounding-box predictions.

### Weaknesses

**W1. Limited tool scope in main paper.** The core experiments focus exclusively on a single zoom-in tool (`image_zoom_in_tool`). While the multi-tool extension in Appendix M is encouraging, it provides limited quantitative evidence (only Table 13 with two rows) and lacks detailed analysis of how the dual-mode decision interacts with different tool types. In particular, the paper does not analyze whether certain tool categories (e.g., perception tools vs. logical/computational tools) exhibit different optimal invocation patterns. Promoting key multi-tool results to the main paper would significantly strengthen the contribution.

**W2. Moderate originality relative to concurrent work.** The concept of adaptive tool invocation is well-established in NLP (Toolformer, ReAct). In the multimodal domain, concurrent works like OpenThinkIMG (Su et al., 2025b) already explore RL-based adaptive tool strategies, and MeCo (in text LLMs) uses meta-cognitive scores to decide tool necessity. AutoTool's primary novelty lies in the specific combination of MSPO's asymmetric penalty and AMB's frequency-based balancing — these are effective engineering contributions but represent incremental rather than conceptual advances over the existing RL tool-use paradigm. The paper would benefit from a more explicit positioning against these concurrent methods, ideally with direct experimental comparisons where feasible.

**W3. Reward model dependency and reproducibility concerns.** The training pipeline relies on Qwen2.5-72B-Instruct as an online judge for open-ended accuracy evaluation, requiring 2 additional H200 GPUs. Appendix J shows that downgrading to a 7B reward model causes V\* overall to drop from 90.1% to 80.1% — a 10-point degradation that reveals a strong coupling between policy performance and judge capability. The main paper does not disclose the reward model's prompt template, temperature, or sampling settings (partially addressed in Appendix B), nor does it report the judge's agreement rate with human annotations. This limits external reproducibility and makes it difficult to disentangle AutoTool's algorithmic contribution from the reward model's quality.

**W4. Sequential tool chains remain unaddressed.** The paper acknowledges (Appendix O) that identifying redundant steps within multi-step tool chains is an open challenge. The current MSPO framework only evaluates the final answer, which means a trajectory that wastefully invokes tools multiple times but eventually produces a correct answer receives the same reward as an efficient one-shot trajectory. This is a meaningful limitation for scaling to more complex agentic tasks where tool chains are the norm.

---

> ### Author Rebuttal · Authors · 2026-03-30
>
> Thank you for your positive assessment of our work. We address your comments in detail below and the changes we will make to the paper.
>
> > Q1/W3. Reward model sensitivity and reproducibility
>
> We will provide the full reward model prompt template in the revision. Due to space limitations, we do not include the complete prompt here. In summary, the judge is instructed to compare a model-generated answer with a standard answer and output a binary consistency label (1/0) based on semantic equivalence, with few-shot examples. For sampling, the reward model is deployed via vLLM with temperature set to 0.3, without top-p or top-k truncation, and other settings follow default configurations.
>
> To evaluate alignment with human judgment, we randomly sample 200 cases and compare the judge outputs with manual annotations.
>
> |Model|Agreement Rate|
> |-|-|
> |Qwen2.5-72B-Instruct|1.000|
> |Qwen2.5-32B-Instruct|0.985|
> |Qwen2.5-14B-Instruct|0.980|
> |Qwen2.5-7B-Instruct|0.940|
> |LLaMA-3-70B-Instruct|0.985|
> |LLaMA-3-8B-Instruct|0.935|
>
> Agreement with human annotations is high across models, with Qwen2.5-72B reaching 1.000, while smaller models exhibit slightly lower agreement.
>
> We also test different reward models (*e.g.*, LLaMA-3 series) and observe similar trends, indicating robustness across reward models.
>
> |Reward Model|HRbench-4K|HRbench-8K|V*|
> |-|-|-|-|
> |LLaMA-3-8B|72.8|67.9|80.6|
> |LLaMA-3-70B|76.4|73.5|89.0|
> |Qwen2.5-72B|76.9|74.0|90.1|
>
> > Q2. Mode Decision Quality Evaluation
>
> To assess mode selection quality, we conduct controlled experiments with forced reasoning modes (see Table 12 in Appendix K). Furthermore, we analyze cases where changing the reasoning mode leads to different outcomes.
>
> |Benchmark|#Changed|AutoTool Correct|Acc (%)|
> |-|-|-|-|
> |V\*|17|12|70.6|
> |HRBench|35|28|80.0|
>
> AutoTool selects the better-performing mode in most cases, indicating effective decision alignment.
>
> > Q3/W1. Multi-tool Generalization Depth
>
> In the DeepEyesV2 extension, the dual-mode mechanism remains unchanged: the model predicts `<tool_on>` or `<tool_off>` at the start. If `<tool_on>` is selected, it performs iterative tool-assisted reasoning; otherwise, it relies on internal reasoning.
> The model learns differentiated patterns across tool types. Both perception and computational tools are implemented as executable Python functions, enabling unified interaction with distinct behaviors. We report per-tool usage and accuracy on V* and MathVista.
>
> |Tool Type|V* Ratio (%)|V* Acc (%)|MathVista Ratio (%)|MathVista Acc (%)|
> |-|-|-|-|-|
> |Crop|81.82|85.0|13.40|76.8|
> |Numerical Analysis|0.00|--|80.07|74.7|
> |Mark|17.11|87.5|5.55|76.5|
> |Other|1.07|0.0|0.98|16.7|
>
> Tool usage strongly correlates with task characteristics. On the vision-centric V\*, the model mainly uses perception tools such as cropping, while on the reasoning-intensive MathVista, numerical tools dominate. Tools also exhibit complementary effects (*e.g.*, mark + crop improves accuracy in V\*), whereas less relevant tools (*e.g.*, rotation or image enhancement) degrade performance, indicating that inappropriate usage can hinder reasoning.
>
> > Q4/W4. Handling Sequential Tool Chains.
>
> Trajectory-level rewards is important but difficult to define. The utility of each tool call is hard to quantify, as tools may provide intermediate evidence or verify reasoning, so naive penalties on trajectory length or step-wise rewards may suppress useful exploration.
>
> Despite using only final-answer supervision, the model learns non-trivial tool usage patterns, as shown in **Q3/W1. Multi-tool Generalization Depth**, where tool invocation correlates with task characteristics, and inappropriate usage degrades performance. This suggests implicit utility learning.
>
> Incorporating trajectory-level rewards is a promising direction. Possible approaches include proxy rewards via counterfactual comparisons or soft regularization on redundant usage. We leave a systematic study to future work.
>
> > W2. Moderate originality relative to concurrent work.
>
> We clarify that our primary contribution is not designing a better tool-necessity classifier via direct supervision, but expanding the RL search space through sufficient dual-mode exploration, enabling the model to self-direct appropriate reasoning strategies.
>
> Existing methods such as OpenThinkIMG introduce RL-based adaptive policies without explicitly regulating the exploration of reasoning modes, leading to mode collapse (e.g., tool usage $\sim$0.1). In contrast, AutoTool explicitly enforces balanced exploration during training, preventing suboptimal policies.
>
> Compared to MeCo, which uses a meta-cognitive trigger as a plug-in for tool decisions, our approach improves policy learning through end-to-end training rather than relying on inference-time decision signals. This distinction is important in multimodal settings, where the cost–benefit trade-off of tool use is harder to capture with heuristic or post-hoc rules.

---

### Decision · Program_Chairs · 2026-04-30

**Decision:**

Accept (regular)

**Comment:**

This paper addresses a problem of the current Tool-use learning algorithm -- blindly encouraging tool invocations. It proposes a reinforcement learning framework named AutoTool which can adaptively decide whether to invoke tools on for each query. Experimental results show a significant improvement on V* benchmark compared to the baseline.

Reviewers consider that this is an important problem, the proposed RL framework is well designed, and the improvement on benchmark is significant. Most of questions from reviewers have been addressed in the authors' response.

Based on these comments, the final decision is Accept. The authors should include the responses to reviewers' comments in the camera ready version.